

# Large uncertainty in ecosystem carbon dynamics resulting from ambiguous numerical coupling of carbon and nitrogen biogeochemistry: A demonstration with the ACME land model

Jinyun Tang, William J. Riley

Earth & Environmental Sciences Area, Lawrence Berkeley National Laboratory, Berkeley, 94720, USA

*Correspondence to*: Jinyun Tang (jinyuntang@lbl.gov)

**Abstract.** Most Earth System Models (ESMs) have incorporated, or are incorporating, coupled carbon and nutrient dynamics in their land modules. We show here that different numerical implementations of nutrient controls may imply different ecological mechanisms not recognized in the original model design and can have first order impacts on predicted terrestrial

carbon cycling. Using ALM, the land module of the DOE ESM (ACME), we analysed land-atmosphere $CO_2$ exchange with coupled carbon and nitrogen dynamics through three commonly-applied numerical implementations of nitrogen limitation: (1) Mineral Nitrogen based Limitation (MNL), (2) Net nitrogen Uptake based Limitation (NUL), and (3) Proportional Nitrogen flux based Limitation (PNL). By the last decade of the contemporary period (1850-2000), the three schemes resulted in very similar global terrestrial carbon and nitrogen distributions. However, under an RCP4.5 $CO_2$ concentration

forcing, the implementations resulted in wildly diverging 2001-2300 land-atmosphere $CO_2$ exchanges. Quantitatively, the divergence is as large as that of the CMIP5 models by 2100 and is about 1900 Pg C (~890 ppmv) by 2300. Our analysis suggests that these differences result from: (1) the typically high predicted terrestrial ecosystem carbon to nitrogen ratios (i.e., nutrient constrained conditions) and (2) the schemes predict different levels of nitrogen limitation to the carbon cycle, so that the PNL scheme leads to larger nitrogen loss through aerobic and anaerobic denitrification and surface and subsurface

hydrological transport. We also found significant sensitivity of model predictions to initial conditions and numerical time step size but insignificant sensitivity to the sequence of numerical oxygen and nitrogen limitation or the ordering of reaction and chemical transport. We conclude that inconsistencies in imposing nutrient limitations will very likely lead to large uncertainties in predicted carbon stocks and long-term carbon-climate feedbacks. Finally, we recommend approaches to systematically alleviate these uncertainties.

**Keywords**: carbon-nitrogen feedbacks, nitrogen limitation, land-atmosphere $CO_2$ exchange, RCP4.5, law of the minimum

## 1 Introduction

Earth System Models (ESMs) used for assessing future climate and related processes rely on large-scale land biogeochemical (BGC) models to simulate ecosystem responses to changing atmospheric $CO_2$, temperature, precipitation,



nitrogen (N) deposition, and etc. Recent work analysing ESM land models that participated in the Coupled Model Intercomparison Project Phase 5 (CMIP5) showed very large differences among those models' predictions (e.g., Arora et al., 2013; Friedlingstein et al., 2014; Shao et al., 2013; Koven et al. 2015a). Such differences are often attributed to the four types of uncertainties, including structural (Tang et al., 2010; Wieder et al., 2015a), numerical (Yeh and Tripathi, 1989),

parameterization (Tang and Zhuang, 2008; Luo et al., 2015), and forcing data (Clein et al., 2007; Blanke et al., 2016), which are, respectively, loosely related to the four stages of BGC model design: (I) conceptualizing the relevant mechanisms and translating them into governing equations; (II) numerical encoding of the governing equations; (III) process module calibration and parameterization; and (IV) model analyses and applications. There have been numerous examples of how one could quantify and reduce these uncertainties (e.g., Tang and Zhuang, 2008, 2009; Williams et al., 2009; Lichstein et al.,

2014; Wei et al., 2014; Shi et al., 2015). Here we describe a new type of uncertainty that is a combination of type-I and type-II, and can result in predictions of ecosystem carbon dynamics as divergent as that of CMIP5 land models.

We report our findings using the carbon-nitrogen coupling as an example; however, this new type of uncertainty is related to the broader issue of carbon-nutrient coupling in all kinds of BGC models. Specifically, it relates to how one should numerically represent the fact that different substrates can limit ecosystem biogeochemical processes under different

conditions. For instance, it is believed that many terrestrial ecosystems are nitrogen limited (Vitousek and Howarth, 1991; LeBauer and Treseder, 2008), because breaking down the triple bond of dinitrogen ($N_2$) and converting it into assimilable forms requires a significant fraction of newly assimilated or reserved carbon (Gutschick, 1987). Tropical forest ecosystems are often regarded as phosphorus limited because of their highly weathered soils (Walker and Syers, 1976), but nitrogen or even carbon or potassium limitation can still occur (e.g., Wright et al., 2011; Fanin et al., 2015). In moist environments, such

as wetlands, where organic matter decomposition is more likely oxygen limited, anaerobic decomposition dominates but aerobic decomposition may proceed simultaneously (DeBusk et al., 2001). Given this wide range of substrate limitation conditions, it is therefore logical to ask: how would different numerical treatments of substrate limitation influence the prediction of a land BGC model?

We answer the above question by focusing on nitrogen—the most important macronutrient related to whether or not

terrestrial ecosystems could continue to sequester anthropogenic $CO_2$ (Oren et al., 2001; Drake et al., 2011; Grant, 2013). We note there are two aspects that determine the modelled influence of nitrogen on ecosystem carbon dynamics: (1) the mechanistic formulation of carbon and nitrogen coupling and (2) the numerical implementation of a given formulation. For the first, there are many opinions under debate (e.g., Niu et al., 2016; Stocker et al., 2016), whereas the second has been rarely discussed, and is our focus here.

We begin our analysis with the equation for a generic substrate $S$ in a soil control volume:





$$\frac{dS}{dt} = F_{S,input} - F_{S,uptake} \tag{1}$$

where $F_{S,input}$ and $F_{S,uptake}$ are, respectively, substrate input (from all sources) and substrate uptake (by all competing entities). Here and below, unless otherwise stated explicitly, we assume the units for all variables in a given equation are consistently defined.

When substrate $S$ is mineral nitrogen (we use $S$ and mineral nitrogen interchangeably unless a clarification is

required), input $F_{S,input}$ includes fertilization, atmospheric nitrogen deposition, nitrogen fixation, and microbial nitrogen

mineralization from soil organic matter (SOM) decomposition; while $F_{S,uptake}$ includes plant assimilation and microbial

utilization. If the interaction between soil mineral surfaces and ammonium nitrogen is considered (e.g. Gerber et al., 2010),

$F_{S,input}$ and $F_{S,uptake}$ should be modified accordingly, depending on whether ammonium is adsorbed or desorbed from soil

minerals. To further simplify the problem, we solved the overall spatiotemporal evolution of substrate $S$ (which is a

function of both transport and biogeochemistry) using the operator splitting approach (e.g., Strang, 1968; Tang et al., 2013),

so that $F_{S,input}$ and $F_{S,uptake}$ refer, respectively, to nitrogen mineralization (by decomposers) and nitrogen immobilization

(by microbes and plants).

Equation (1) may be approximated using the forward Euler scheme (e.g. Atkinson, 1989):

$$S(t + \Delta t) = S(t) + \left( F_{S,input} - F_{S,uptake} \right) \Delta t \tag{2}$$

With a given numerical time step $\Delta t$, if $S(t + \Delta t)$ becomes negative (before any adjustment to the rates), the

biogeochemical system is defined as substrate-$S$ limited during that numerical time step. We note that this numerical definition of nitrogen limitation (which operates on time scales from minutes to hours) appears different from the ecological definition, which is defined as stimulated ecosystem productivity in response to nitrogen addition and operates on time scales from days to years (Vitousek and Howarth, 1991). However, in a BGC model, ecological nitrogen limitation is realized as an emergent response accumulated from many within time-step nitrogen limitations.

Using a higher order numerical scheme will not avoid substrate limitation, and, when substrate limitation occurs, the high order scheme will usually become first order (Bolley and Crouzeix, 1978), a result that also holds for implicit schemes (Hundsdorfer and Verwer, 2003). Higher order accuracy may be achieved if both the substrate production and destruction rates are modified simultaneously (e.g., Burchard et al., 2003), but such an approach will fail when substrate production is independent of consumption, a situation that occurs exactly in the CENTURY-like models (Parton et al. 1988;

Koven et al. 2013), where the activity of nitrogen mineralizers is independent from that of nitrogen immobilizers (Tang and Riley, 2016). Nor will an adaptive time stepping approach resolve this numerical substrate limitation problem, because that



would require an impractically small time-step to avoid negative numerical solutions (Formaggia and Scott, 2011). Nevertheless, a numerical nitrogen limitation as applied in equation (2) does depend on the time step size, but as we demonstrate below, this uncertainty is secondary to that from using different numerical implementations of nitrogen limitation.

We now analyse three legitimate and commonly applied numerical methods to resolve substrate limitation when solving equation (2). The three numerical approaches imply different coupling between nitrogen competitors and producers in the model, and therefore lead to different (sometimes unacknowledged) ecological coupling between carbon and nitrogen dynamics.

       The first nitrogen uptake limitation approach has been adopted by models like CLM-CNP (Yang et al., 2014),
BiomeBGC (Thornton et al., 2002), BiomeBGC MuSo (Hidy and Barcza, 2014), CLM4.0 (Oleson et al., 2010), CLM4.5 (Oleson et al., 2013), the to be released CLM5.0, and one version of ALMv1 (the land model in the DOE earth system model ACME-v1):

$$\overline{F}_{S,uptake} = \min\left\{\frac{S(t)/\Delta t}{F_{S,uptake}}, 1\right\} F_{S,uptake} \tag{3}$$

Equation (3) assumes that the actual total nitrogen uptake $\overline{F}_{S,uptake}$ is limited solely by the available mineral nitrogen $S(t)$ and is not affected by mineral nitrogen released from SOM decomposition during the numerical time step, which is certainly
inconsistent with the governing equation. In the following, we name this approach (i.e., equation (3)) as the **M**ineral **N**itrogen based **L**imitation scheme (MNL).

       In some models, like CABLE (Wang et al., 2010) or the Generic Decomposition and Yield model (Comins and McMurtrie, 1993), the $j$-th sub-component $F_{S,uptake,j}$ of $F_{S,uptake}$ may already include substrate limitation based on the availability of $S(t)$. These models apply either equation (3) or its variants (to be introduced later), or a "numerical" Monod
term (e.g., Tang et al., 2016) that has no chemical or biological kinetic meaning (as in contrast to the enzymatic Monod function) to the $j$-th potential uptake flux $F_{0,S,uptake,j}$. When both nitrogen and phosphorus are considered for an entity of fixed stoichiometry (e.g., a decomposing organic matter pool or a microbe), the imposition of substrate limitation is even more uncertain. One approach is to use the potential nitrogen uptake flux $F_{0,N,uptake}$ and phosphorus uptake flux $F_{0,P,uptake}$ to first calculate the nitrogen-limiting factor $x_N$ and phosphorus-limiting factor $x_P$

$$x_N = \min\left\{\frac{MIN_N/\Delta t}{F_{0,N,uptake}}, 1\right\} \tag{4}$$



$$x_P = \min\left\{\frac{MIN_P/\Delta t}{F_{0,P,uptake}}, 1\right\} \tag{5}$$

Then Liebig's law of the minimum is applied by taking the minimum of $x_N$ and $x_P$ to compute an overall limiting factor $x_{NP}$ that constrains the overall decomposition flux, which by stoichiometry balance will lead to down-regulated nitrogen and phosphorus uptake rates $F_{N,uptake,j}$ and $F_{P,uptake,j}$ that are then used to resolve the nitrogen and phosphorus competition.

Occasionally, $x_N$ and $x_P$ may be calculated using Monod functions for each of the substrate competing entities, leading to

a premature application of the law of the minimum (e.g., Leon and Tumpson, 1975; Danger et al., 2008). However (as we explained in Appendix A), such application of the 'law of the minimum' mistakes the system-wise nutrient limitation as a local constraint on subcomponents of the system. Given the limited amount of nitrogen and phosphorus available for competition, an additional application of either equation (3) (or the to be introduced equations (6) or (7)) may still be imposed to avoid negative numerical solutions when all competing fluxes are resolved (i.e., a second application of the law

of the minimum will be introduced automatically through the numerical integration). Such a strategy then leads to a double counting of nutrient limitation if the mass balance is imposed strictly, and if the mass balance is not imposed strictly, an unwanted numerical nutrient fertilization might occur (e.g., the ODE45 solver as demonstrated in Tang and Riley (2016)).

The second nitrogen limitations scheme that we analyse here is represented as:

$$\overline{F}_{S,uptake} = \min\left\{\frac{S(t)/\Delta t}{F_{S,uptake} - F_{S,input}}, 1\right\} F_{S,uptake} \tag{6}$$

We name equation (6) the **N**et nitrogen **U**ptake based **L**imitation (NUL) scheme (note when NUL is applied, it holds that

$F_{S,uptake} - F_{S,input} > 0$). The NUL scheme is based on the approach of derivative clipping, and is used in MATLAB's ODE45 (Shampine et al., 2005). However, ODE45 imposes equation (6) by violating the law of mass balance (Tang and Riley, 2016). We avoid this problem here by applying the flux adjustment only to $F_{S,uptake}$, because $F_{S,input}$ is assumed independent from substrate $S$ in equation (6).

The third nitrogen limitation scheme is

$$\overline{F}_{S,uptake} = \min\left\{\frac{F_{S,input} + S(t)/\Delta t}{F_{S,uptake}}, 1\right\} F_{S,uptake} \tag{7}$$

We name equation (7) as the **P**roportional **N**itrogen flux based **L**imitation (PNL), and it is the only numerical scheme (among the three we analysed) that is consistent with the governing equation. PNL assumes that the newly (i.e., within the



time step) released mineral nitrogen ($F_{S,input}$) and existing (i.e., at the beginning of the time step) mineral nitrogen $S(t)$ are equally accessible to immobilizers. This assumption is an oversimplification because diffusion can limit the newly released and existing mineral nitrogen from mixing completely in the soil over the typically short time steps in land models (0.5 – 1 h) (Schimel and Bennett, 2004). Therefore PNL will underestimate the true nitrogen limitation. We also note that when

diffusion limitation is ignored, assuming whether or not plants and microbes have absolute priority of newly released mineral nitrogen over existing mineral nitrogen will not change the form of equation (7). A modified PNL scheme that includes diffusion constraints is used in the *ecosys* model (e.g. Grant, 2013) to rectify overly large nutrient uptake fluxes (personal discussion with R. Grant, 2016) that can lead to negative nutrient concentrations.

        There have been other schemes proposed for nitrogen limitation (which however will not be analysed in this study).

For instance, Wang et al. (2010) in their constraint of decomposition due to nitrogen limitation (cf. their equation C12) calculated the de facto decomposer nitrogen uptake as

$$\bar{F} = \min\left\{ \max\left\{ 1 + \frac{\left( F_{S,input} - F_{S,uptake} \right)\Delta t}{S(t)}, 0 \right\}, 1 \right\} F_{S,uptake} \qquad (8)$$

where $F_{S,input}$ and $F_{S,uptake}$ refer, respectively, to nitrogen mineralization and microbial nitrogen immobilization. Equation (8) reduces nitrogen uptake when the net mineralization $F_{S,input} - F_{S,uptake}$ is negative, and has no effect on nitrogen uptake when net mineralization is positive (we acknowledge that equation (8) is a more complete form to their equation C12

because their equation C12 was only applied to negative net nitrogen mineralization). But as we explained above, this approach will not avoid predicted negative nitrogen concentrations and further adjustments as represented in the MNL, NUL or PNL scheme are needed.

        Numerically, MNL, NUL, and PNL are all legitimate approximations to the same governing equation (1) as discretized in the forward Euler form equation (2) (type-II uncertainty). They nevertheless represent different

biogeochemical coupling between mineral nitrogen, plants, and microbes (type-I uncertainty). When the actual numerical representation of nitrogen limitation is not explicitly reported (which is common in the literature), one would regard the BGC models using these three schemes as structurally identical and numerically similar (and indeed for non-nitrogen limited conditions, these three approaches lead to identical model predictions (Tang and Riley, 2016)). However, because ecosystem carbon sequestration is the difference between several large magnitude nitrogen-limited ecosystem carbon fluxes, we

hypothesize that different nitrogen limitation schemes will lead to different predictions of ecosystem carbon dynamics.

    We therefore address two hypotheses:

    (H1): The ambiguous numerical implementation of nitrogen limitation leads to large uncertainty in simulated ecosystem carbon dynamics.





(H2): Uncertainty from the model time-step size is smaller than that resulting from the use of different nutrient limitation schemes.

We evaluated the above hypotheses using the ALMv1 model that integrates BeTR—a numerically robust reactive transport module (RTM) for biogeochemical transport and reactions (Tang et al., 2013)—with simulations of both historical

and future RCP4.5 emission scenario atmospheric $CO_2$ forcing. Below we describe the model configurations and simulation protocols, present and discuss our model results, and finally give recommendations on how to remove or alleviate this new type of uncertainty (i.e., ambiguous numerical nutrient limitation).

## 2. Methods

### 2.1 Model configuration

We applied ALMv1-BeTR to explore how different numerical schemes of nitrogen limitation affect the predicted ecosystem uptake of atmospheric $CO_2$. BeTR is a multiphase RTM that consistently represents the transport (including multiphase diffusion, advection, ebullition, and gas phase arenchyma transport) for an arbitrary number of chemical tracers, which for this study includes seven carbon pools (Koven et al., 2013), and eight abiotic tracers, $N_2$, $O_2$, Ar, $CH_4$, $CO_2$, $NH_4^+$, $N_2O$, and $NO_3^-$. Compared to the first version of BeTR in CLM4 (Tang et al., 2013), ALMv1-BeTR improves the numerical

treatment of dual phase diffusion (Tang and Riley, 2014) and advection (Manson and Wallis, 2000) (see Figure S1 for a demonstration of its numerical accuracy in tracer transport), and uses F90's object oriented polymorphism to implement different BGC formulations within the same biophysical environment. As in the default ALMv1 BGC, which is the de facto CLM4.5BGC (Koven et al., 2013; Oleson et al., 2013), all BeTR BGC implementations do not physically transport $NH_4^+$.

We implemented the biogeochemistry of all BeTR BGC models using the Peterson matrix based formulation (Russell, 2006;

Tang and Riley, 2016), so that minimal modification was needed to implement the three nitrogen limitation schemes (i.e., MNL, NUL, and PNL).

We note that CLM4.5BGC uses an instantaneous, relative demand (RD) down-regulation scheme for GPP. Under nitrogen-limited conditions, this RD down-regulation scheme first calculates the ratio between existing soil mineral nitrogen pool and total potential nitrogen uptake (by plants and microbes) to avoid negative mineral nitrogen stock, and then

multiplies this ratio with the nitrogen unlimited GPP to obtain the down-regulated GPP. This RD approach unrealistically assumes that root nutrient uptake instantaneously affects leaf photosynthesis and artificially restricts the plant and microbial nutrient competition to occur before plant carbon allocation. We recently showed that this approach (1) leads to very unrealistic diurnal GPP cycles (Ghimire et al. 2016) and (2) has not been corroborated by observations (Zhu et al., 2016), even though it may be ecologically convenient for analysing long-term ecosystem biogeochemistry with a time step of years.

We therefore removed this down-regulation scheme in all BeTR BGC simulations.



CLM4.5 and ALMv1 employ a CENTURY-like (Parton et al., 1988) soil BGC formulation, which represents microbial population dynamics and associated biogeochemical activity implicitly. The model allows plants and microbes to compete equally for $NH_4^+$ and $NO_3^-$, and assumes that both plants and organic matter decomposers assimilate $NH_4^+$ over $NO_3^-$. The first assumption is under intense debate (e.g., Gerber et al., 2010; Zaehle and Friend, 2010; Thomas et al., 2015;

Niu et al., 2016; Zhu et al., 2016), whereas the second assumption is very likely unrealistic because (1) it restricts the model to execute nitrogen limitation after oxygen limitation (as $NO_3^-$ demand by denitrifiers is a function of oxygen and applying nitrogen limitation requires knowing the relative uptake demand of $NH_4^+$ over $NO_3^-$), even though they occur simultaneously in the real world and (2) a grid cell in any large scale BGC model actually represents the average of a heterogeneous soil, so the uptake of $NO_3^-$ should never be zero as long as some $NO_3^-$ exists.

To evaluate hypothesis (H1), we used five BGC model configurations implemented in ALMv1-BeTR (Table 1). Among them, the three BGC formulations (MNL, NUL, and PNL) differ in their numerical interpretations of nitrogen limitation. Since all model configurations in BeTR require identical model inputs, we also tested the model sensitivity to initial conditions by comparing PNL with PNLIC, where the latter uses the code base of PNL and initial conditions from the NUL simulation. Simulations PNLIC and NUL are compared to demonstrate the effect of different nitrogen limitation

implementations with the same initial conditions. The final model configuration, PNLO, when compared to PNL, illustrates the ordering dependence of substrate limitation (for oxygen and nitrogen). To make the model PNLO run, we first predicted the relative demand for $NH_4^+$ and $NO_3^-$ based on total mineral nitrogen availability, then implemented oxygen limitation on nitrification and decomposition, and finally applied nitrogen limitation to microbes and plants a second time. This requirement for applying nitrogen limitation twice is not easily observable from the governing equations of the BGC model

and demonstrates (1) that the default ALM BGC and CLM4.5 BGC model structures of plant-soil nitrogen interactions are problematic and (2) (once more) that numerical implementations of nutrient limitations in ESM land models imply (sometimes unacknowledged) different ecological dynamics that is not described in the governing equations.

The second hypothesis (H2) is evaluated with four example single gridcell simulations in geographically and climatically distinct locations (Figure 3): (74.67°W, 40.6°N; Eastern U.S.), (26.22°E, 67.7°N; Northern Finland), (50.02°W,

4.88°S; North East Brazil), and (51.5°W, 30.0°S; South Brazil). These gridcells were chosen to illustrate spatial heterogeneity in how time stepping strategies would influence simulated ecosystem carbon dynamics. We adopted the strategy from Tang and Riley (2016) (their appendix D) for adaptive time stepping and designated relevant simulations with PNL-adapt.



**2.2 Simulation protocol**

All model simulations were first run to preindustrial equilibrium using the spinup protocol in Koven et al. (2013) with the QIAN climate forcing data (cycled for 1948-1972; Qian et al., 2006). The model output by the end of spinup was then used for simulations in the contemporary period 1850-2000 with diagnostic atmospheric $CO_2$ concentrations. The

RCP4.5 scenario atmospheric $CO_2$ concentrations (starting from 2006) were used together with the QIAN climate forcing for the simulation period 2001-2300 (see Figure S2b). We did not apply the climate anomaly representing future climate change to the RCP simulations; therefore the simulated carbon dynamics over 2001-2300 only represented the effects of changing atmospheric nitrogen deposition (Figure S2a), atmospheric $CO_2$ (Figure S2b), and land use change. We expect that including more uncertainty sources (such as uncertain future climate) will further strengthen the conclusions of our study.

**3 Results**

**3.1 Global simulations for the contemporary period 1850-2000**

For the last decade (1991-2000) of the historical simulation period 1850-2000, the six model simulations gave very similar latitudinal distributions of several important variables (Figure 1). Small differences were found for latitudinal distributions of total soil organic carbon (Figure 1f), total soil organic nitrogen (Figure 1g), total vegetation carbon (Figure

1h) and total vegetation nitrogen (Figure 1i). Particularly for the July latent heat flux (Figure 1e), all simulations overlap, which is consistent with the relatively small differences in July leaf area index (LAI), GPP, NPP, and total vegetation carbon (Figure 1a, b, c, and h) and that plant transpiration has a fast response to climate forcing (which is the same in all six model simulations). The overall close agreement between the default simulation (purple line) and all five BeTR-based simulations indicates that it requires a long time for the effect of different nitrogen limitation schemes to emerge in the simulations, an

observation that is consistent with the usually high ecosystem carbon to nitrogen ratio and that ecosystem carbon stocks are cumulative differences between the large fluxes of ecosystem carbon uptake (i.e., GPP) and ecosystem carbon loss (respiration and disturbances).

In contrast to the high degree of similarity between many of the variables simulated by the five BeTR-based models, the historical trajectories of cumulative NEE (positive means emitting $CO_2$ into the atmosphere) are very different

(Figure 1a). Among the MNL, NUL, and PNL simulations PNL (red line) had higher land carbon release compared to NUL (green line; an almost carbon neutral land by year 2000) and MNL (blue line; a cumulative land carbon uptake of about 40 Pg C by year 2000). The cumulative NEE simulated by PNLO (black line) is very similar to that by PNL, yet the ordering dependence still lowered the cumulative carbon release by about 50 Pg C compared to PNL by year 2000. PNLIC (cyan line) showed an anomalously high release of land carbon resulting from enhanced decomposition of coarse woody debris (Figures

S3-a2, b2, and c2), which is reflected in the higher heterotrophic respiration (Figure S4c) driven by more efficient decomposer nitrogen immobilization in PNL as compared to NUL. Although an in depth analysis will be provided using point simulations (section 3.3), this more efficient nitrogen uptake can be simply explained by observing the similar nitrogen





input from fixation and deposition between the models (results are not shown but can be inferred from the almost overlapping NPP, which controls nitrogen fixation in this version of ALM), and (as shown in Supplemental Material) that the nitrogen uptake calculated from equation (6) (for NUL) is smaller than that from equation (7) (for PNL). The huge difference between PNL and PNLIC in the cumulative NEE (Figure 1a) indicates that the CENTURY-like BGC model is

very sensitive to initial conditions, corroborating the finding in Exbrayat et al. (2014). Finally, we observed very small differences in the latitudinal distributions of soil mineral nitrogen over 1991-2000 between the five BeTR-based simulations, and those concentrations are lower than that simulated by the default model (Figure S5).

**3.2 Global simulations for the period 2001-2300**

Although having very similar carbon and nitrogen stocks for the decade of 1991-2000 (Figure 1), the five BeTR
simulations driven by the RCP4.5 atmospheric $CO_2$ concentrations diverged into three groups for 2001-2300 (Figure 2). For the north temperate region (i.e., north of 23.2° N and south of 66.3° N; Figure 2a1), simulations NUL and MNL almost overlapped and predicted a carbon gain of about 42 Pg C (~5600 g C m$^{-2}$) by year 2300; simulations PNLO and PNL almost overlapped and predicted a carbon gain of about 10 Pg C (~1300 g C m$^{-2}$) by year 2300; and PNLIC predicted a small carbon loss of about −12 Pg C (~ −1600 g C m$^{-2}$) by year 2300. The tropics (defined as the region between 23.2° S and 23.2° N)
showed larger divergence (Figure 2b1) with a high carbon gain predicted by MNL about 380 Pg C (~8800 g C m$^{-2}$) and NUL about 360 Pg C (~8300 g C m$^{-2}$) by year 2300, and a lower carbon gain by PNL about 160 Pg C (~3700 g C m$^{-2}$) and PNLO about 140 Pg C (~3200 g C m$^{-2}$), and about −210 Pg C (~4900 g C m$^{-2}$) loss by PNLIC. The divergence in the Arctic region (defined with latitudes north of 66.3° N; Figure 2c1) is comparable to (or, based on per unit area, is higher than) the north temperate region, with a high carbon gain about 16 Pg C (~ 3500 g C m$^{-2}$ by MNL and NUL), a small carbon loss about −4
Pg C (~ −880 g C m$^{-2}$ by PNL and PNLO), and a large carbon loss about −42 Pg C (~ −9200 g C m$^{-2}$) by year 2300.

While we do not place high confidence on the predicted numerical value (as discussed previously and below), the global terrestrial carbon stocks change (Figure S6) between 2006 and 2100 spreads from −5 Pg C (weak source; PNLIC) to 710 Pg C (sink; MNL), which approximately encapsulates the range reported in Shao et al. (2013) (their Table 4; excluding GFDL-ESM2M and MPI-ESM-LR) for the CMIP5 simulations. By 2300, the predicted global terrestrial carbon stock
change ranges from a source of about −400 Pg C (PNLIC) to a carbon sink of about 1500 Pg C (with MNL being slightly higher than NUL). We note that this 1500 Pg C sink is close to a reduction of 700 ppmv atmospheric $CO_2$ which is greater than the 550 ppmv atmospheric $CO_2$ forcing. This clearly indicates that the BGC model structure is questionable (and so we don't place a good confidence on this numbers). Terrestrial carbon stock changes for the PNL and PNLO simulations fall between the predictions by PNLIC and MNL, with a carbon sink of 520 and 470 Pg C, respectively (Figure S6a). Since
nitrogen limitation quantitatively increases across the model configurations (PNL < NUL < MNL), the sequential increases in carbon uptake (MNL > NUL > PNL) in response to the RCP4.5 atmospheric $CO_2$ trajectory imply that ALM-v0 and CLM4.5 (which both use the MNL scheme) may predict too strong global $CO_2$ and nitrogen fertilization effects. We acknowledge that this stronger $CO_2$ fertilization effect resulting from stronger nitrogen limitation (as implied in the



numerical implementations; see Supplemental Material) may first appear counter-intuitive, yet it can be reasonably explained through relevant ecological mechanisms (see discussion in section 4.1). We also found that the predicted total soil carbon change is more sensitive than the total vegetation carbon change (Figure 2 and Figure S6) in response to the different nitrogen limitation implementations, indicating stronger nitrogen regulation of soil carbon stocks.

### 3.3 Point simulations for the four sites

For the group of simulations conducted at the four grid points (Figure 3), we observed similar divergences as those in the global simulations (Figure 2): the MNL scheme (magenta lines) predicted higher carbon gain than did the PNL scheme (green lines), yet the NUL predictions (black lines) almost overlapped with those by MNL. Invoking adaptive time-stepping (PNL-adapt; blue lines) further decreased the predicted carbon gain, which could be explained by the even more effective nitrogen uptake implied by the PNL scheme under smaller time steps. We also switched the computing order between reaction and transport for PNL-adapt (which like all simulations reported in this text calculates biogeochemical reaction before transport) and only found negligible difference (Figure S7).

### 3.4 Results of hypotheses evaluation

Taking all simulations together, we conclude that hypothesis H1 is affirmed. H2 is satisfied in some, but not other, sites and that the size of the numerical time step could have either significant (Figure 3a) or secondary (Figure 3b, c, and d) importance on simulated ecosystem carbon stocks trajectories.

### 4. Discussion

### 4.1 Reasons for the large C cycle differences between different nitrogen limitation implementations

We observed that PNL, NUL, and MNL schemes predict sequentially stronger nitrogen limitation under the same mineral nitrogen availability (Supplemental Material). For biogeochemical models like ALM that resolve mineral nitrogen into ammonium and nitrate (together with the assumed preference of ammonium over nitrate), this order of limitation translates into sequentially less effective plant and microbial assimilation of ammonium and stronger uptake of nitrate nitrogen. Indeed, we found PNL-adapt predicted the highest nitrification rate (as nitrifiers are competing for ammonium in ALM) followed by PNL and MNL (which overlapped with NUL; see Figure 4a1, b1, c1, a2, b2 and c2), leading to the same ranking of soil nitrate abundance (Figure S8) and nitrate loss through aqueous transport (Figure 4a4, b4, and c4). The difference in denitrification rates as simulated by different nitrogen limitation schemes is also evident, with the lowest value predicted by PNL-adapt, and increasing in MNL (which overlaps NUL) and then PNL. The simulations at 51.5° W, 30.0 ° S (which ALM identifies as a $C_3$ grassland) only qualitatively resemble those at the other three sites, yet the ranking of soil nitrate abundance remains (Figure S8d). Corresponding to the nitrogen dynamics, the ecosystem heterotrophic respiration




also increases in the order of MNL (which overlaps NUL), PNL, and then PNL-adapt, except for the period after 1980 for the fourth site (Figure 5d), indicating a strong sensitivity of carbon dynamics to nitrogen processes.

### 4.2 Strategies for robust carbon and nitrogen coupling

These results show that ambiguous numerical implementation of nitrogen limitation leads to a large carbon cycle

prediction uncertainty. To rectify this situation, we have four recommendations to help achieve a numerically robust coupling between carbon and nitrogen (or more generally nutrient) dynamics.

First and foremost, nutrient limitations should be handled automatically through a robust numerical solver, rather than being applied to individual processes through the convenient law of the minimum, an approach that has yet been challenged by observations (e.g. O'Neill, 1989; Danger et al., 2008), and is mechanistically redundant (appendix A). In

reality, nutrient limitations emerge from continuous interactions among all entities and substrates in the ecosystem. Analytically applying law of the minimum to each of the modelled entities can turn the emergent limitation into a specific mechanism constraint that ignores interactions between competing entities. If a strategy is also employed to avoid (unphysical) negative numerical solutions (which is necessary), an unwanted double counting of substrate limitation will occur. Likewise, the numerical Monod-term based approach (e.g., Tang et al., 2016) incorrectly applies the nutrient

limitation as an emergent constraint, as it introduces a specific mechanism constraint that depends on an ambiguously defined residual substrate concentration.

Second, we recommend models explicitly represent substrate kinetics for substrate competition between all consumers. On the one hand, substrate kinetics naturally have the property that as substrate concentrations decrease, uptake fluxes will smoothly decrease. On the other hand, unlike the numerical Monod term (in Tang et al., 2016; which can be

equally replaced with functions like $S^n/(S^n + K_S)$, where $n$ is the quantitative order and $K_S$ is the numerical half saturation constant), appropriately applied substrate kinetics, e.g., the Equilibrium Chemistry Approximation (ECA) kinetics (Tang and Riley, 2013; Tang, 2015; Zhu et al., 2016), have a mechanistic underpinning for the interactions between entities involved in substrate dynamics. In particular, the ECA kinetics allows for an explicit formulation of entity interactions for each substrate, whereas the application of Michaelis-Menten kinetics will render representation of competitive pressures into

the system-wise numerical constraint, possibly causing inconsistencies between the conceptual model, its governing equations, and the numerical solution (see the litter decomposition example in Tang and Riley (2013)).

Third, we contend that more robust numerical solvers should be employed to solve the BGC governing equations. Terrestrial biogeochemical modelling has traditionally not paid sufficient attention to this issue: model equations are often integrated with the single step Euler forward scheme (with a few exceptions such as the TEM model (Raich et al., 1992) and

the ED model (Knox, 2012), which used multi-step methods such as the Runge-Kutta scheme), and ad hoc manipulations are invoked to rectify the unphysical numerical solutions (e.g., see discussions in Tang and Riley (2016)). This may not be a severe issue when the models are of low complexity (e.g., the CMIP5 models are mostly carbon-only models), where





chances of unphysical solutions are less likely to occur. However, there are urgent scientific reasons to introduce more mechanisms into terrestrial biogeochemical models (e.g., Wieder et al., 2015b) for better and more comprehensive analyses of carbon-climate feedbacks. In particular, the migration from single-layer to vertically resolved models is required to correctly simulate global soil carbon stocks, especially for Arctic ecosystems (Koven et al., 2013, 2015b). For ecosystems

such as peatlands, wetlands, rice paddies and tropical forests, the soil physical environmental will frequently fluctuate between wet and dry conditions, causing strong shifts in soil redox status. These dynamics will make the problem of substrate limitation more likely for different substrates over time. The first order explicit reaction-based flux back tracing algorithm proposed in Tang and Riley (2016) is helpful to avoid unphysical negative substrate concentrations during model execution and is numerically consistent with the processes represented in the governing equations (thus it satisfies the Lax

equivalence theorem (Lax and Richtmyer, 1956)). However, its explicit time stepping approach may cause a 'zigzag' phenomena or premature convergence in some unusual cases (e.g., Figure S9). The implicit scheme may also have strong time stepping dependence, and for complex biogeochemical systems, clipping and variable transform may be needed for the implicit scheme to maintain positive solutions for concentrations (Tang et al., 2016). However, as we discussed, the clipping approaches can introduce mass balance errors into the model. One possible candidate to alleviate the time-stepping

dependence is the exponential integrator (e.g. Tuckmantel, 2010), but it may still suffer from violating the strict mass balance constraint that is guaranteed in Tang and Riley's approach. We will present our exercise of the exponential integrator elsewhere.

Finally, we suggest that biogeochemical models should provide more transparent methods description for users to identify uncertainties, and then apply approaches to robustly test model structural uncertainty. In reviewing the literature, we

rarely found sufficient information regarding how substrate limitation is numerically implemented in different models. Even when it is available, this information is usually buried within lengthy derivations of the governing equations, making it difficult to determine to what extent the numerical solutions are robust to the types of problems identified above. It is possible to organize a model's governing equations into a set of clearly stated differential and algebraic equations, and solve them by simply invoking available numerical solvers. Such an approach will allow (1) a robust testing of how a model's

simulation depends on the numerical solver and (2) for assessment of model structural uncertainty if multiple models (or model realizations) are solved with the same robust numerical solver. Standardizing this component of land models could dramatically improve prediction uncertainty quantification and facilitate evaluation of new processes, leading to improved analysis of ecosystem dynamics and C-climate interactions.

## 5. Conclusions

The problems associated with ambiguous numerical implementation of substrate limitations are likely present in most ESM land models. Here, we used the coupled carbon-nitrogen dynamics in the ACME land model as an example and demonstrated that the ambiguous numerical implementation of substrate limitation is a serious type of carbon cycle uncertainty, comparable to the uncertainty across the suite of CMIP5 simulations. In particular, such uncertainty may imply



the models are simulating (unacknowledged) ecological mechanisms that are inconsistent with the governing equations, which further lead to uncertainties with initial conditions, and ordering of model integrations. Given that more nutrient mechanisms will be introduced in the next generation of land biogeochemical models, this ambiguity will be even more important and potentially a very large source of uncertainty. For a robust numerical coupling of carbon and nutrient

dynamics, we suggest modellers should: (1) abandon the law of minimum as an analytically explicit constraint to individual entities in the biogeochemical systems; (2) represent substrate competition in their models with explicit substrate kinetics, (3) use more advanced numerical solvers, and (4) document their model implementations with more technique details. With such, we could thence better understand if we are increasing the model complexity for the right reasons.

**Appendix A. Example misuse of Liebig's law of the minimum**

We build our example based on the classic model by Leon and Tumpson (1975), which is

$$\frac{dN_i}{dt} = N_i \left[ \min_j \left\{ \frac{g_{ij}(R_j)}{q_{ij}} \right\} - D_i \right], \begin{cases} i = 1, \cdots, n \\ j = 1, \cdots, n \end{cases} \tag{A-1}$$

$$\frac{dR_i}{dt} = f_j(R_j) - \sum_i q_{ij} \left[ \min_j \left\{ \frac{g_{ij}(R_j)}{q_{ij}} \right\} \right] N_i, \begin{cases} i = 1, \cdots, n \\ j = 1, \cdots, n \end{cases} \tag{A-2}$$

where $N_i$ is consumer $i$ biomass density. $R_j$ is resource $j$ (bio)mass density or concentration, or whatever variable is

appropriate to the form of the resource. $f_j(R_j)$ is net supply rate of resource $j$, which could be either positive or negative.

$g_{ij}(R_j)$ is rate of removal of the $j$-th resource by each individual of the $i$-th consumer population. $q_{ij}$ is the conversion

factor of units of $j$ into units of $i$ (or the reciprocal of substrate use efficiency of $j$-th substrate by $i$-th consumer population).

This model describes the growth of a community of populations (denoted by $i$) on a set of perfectly complementary substrates (denoted by $j$) based on Liebig's law of the minimum. However, this application of law of the minimum is incorrect. We back up our assertion with the following explanation.

Suppose there is only one population feeding on two perfectly complementary substrates, then by approximating equations (A-1) and (A-2) with the Euler forward form, we obtain

$$N_1(t + \Delta t) = N_1(t) + \Delta t N_1(t) \left[ \min \left\{ \frac{g_{11}(R_1(t))}{q_{11}}, \frac{g_{12}(R_2(t))}{q_{12}} \right\} - D_1 \right] \tag{A-3}$$





$$R_1(t+\Delta t) = R_1(t) + \Delta t f_1(R_1) - \Delta t q_{11}\left[\min_j\left\{\frac{g_{ij}(R_j)}{q_{ij}}\right\}\right]N_1, j=1,2 \qquad (A\text{-}4)$$

$$R_2(t+\Delta t) = R_2(t) + \Delta t f_2(R_2) - \Delta t q_{12}\left[\min_j\left\{\frac{g_{ij}(R_j)}{q_{ij}}\right\}\right]N_1, j=1,2 \qquad (A\text{-}5)$$

Now suppose population $N_1$ is locally limited by substrate $R_1$, such that $g_{11}(R_1(t))/q_{11} < g_{12}(R_2(t))/q_{12}$, which leads to

$$N_1(t+\Delta t) = N_1(t) + \Delta t N_1(t)\left[\frac{g_{11}(R_1(t))}{q_{11}} - D_1\right] \qquad (A\text{-}6)$$

$$R_1(t+\Delta t) = R_1(t) + \Delta t\left[f_1(R_1) - g_{11}(R_1)N_1\right] \qquad (A\text{-}7)$$

$$R_2(t+\Delta t) = R_2(t) + \Delta t\left[f_2(R_2) - \frac{q_{12}}{q_{11}}g_{11}(R_1)N_1\right] \qquad (A\text{-}8)$$

Now define

$$\lambda = \frac{g_{11}(R_1(t))}{g_{12}(R_2(t))}\frac{q_{12}}{q_{11}} \qquad (A\text{-}9)$$

Where it can be verified that $\lambda < 1$. Then by entering equation (A-9) into (A-8), we obtain

$$R_2(t+\Delta t) = R_2(t) + \Delta t\left[f_2(R_2) - \lambda g_{12}(R_2)N_1\right] \qquad (A\text{-}10)$$

5    Now if $R_1(t+\Delta t) > 0$ and $R_2(t+\Delta t) < 0$, both of which can be easily satisfied (note $f_2(R_2)$ could be negative), then population $N_1$ is de facto limited by substrate $R_2$, which is opposite to the "local constraint" that the growth of population $N_1$ is limited by substrate $R_1$. Now in order to avoid $R_2(t+\Delta t) < 0$, a numerical substrate limitation must be done, and the use of Liebig's law of minimum in growth rate calculation in equation (A-3) is inappropriate such that it results in a double counting of substrate limitation. For a community of many populations and substrate, we expect such misuse of

10   Liebig's law of minimum could occur even more frequently, and should be avoided.



**Author Contributions**

J.Y. Tang designed the study and conducted the experiments. J.Y. Tang and W. J. Riley discussed the results and wrote the paper.

**Acknowledgments**

This research was supported by the Director, Office of Science, Office of Biological and Environmental Research of the US Department of Energy under contract No. DE-AC02-05CH11231 as part of the Accelerated Climate Model for Energy in the

Earth system Modeling program, as well as the Regional and Global Climate Modeling (RGCM) Program. The study used the Lawrencium computational cluster resource provided by the IT Division at the Lawrence Berkeley National Laboratory. The data used in this paper can be obtained by contacting the first author at jinyuntang@lbl.gov.

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

30





Table 1. Model configurations used to evaluate the uncertainty of ambiguous numerical implementation of nutrient limitation.

| Simulation ID | Model configuration |
| --- | --- |
| MNL | Mineral Nitrogen based Limitation scheme: only existing mineral nitrogen is available for uptake at current time step. It implements equation (3). |
| NUL | Net nitrogen Uptake based Limitation scheme: mineral nitrogen demand is calculated as the residual between total nitrogen demand and gross mineralization. It implements equation (6). |
| PNL | Proportional Nitrogen flux based limitation scheme: mineral nitrogen from gross mineralization and existing soil mineral nitrogen are competed equally by plants and microbes. It implements equation (7). |
| PNLIC | Like PNL, but it uses initial condition from NUL. |
| PNLO | Like PNL, but $O_2$ limitation comes after nitrogen limitation. However, a second nitrogen limitation required for avoiding model crash. |
| Default | ALM-v0, which is the de facto CLM4.5BGC. |





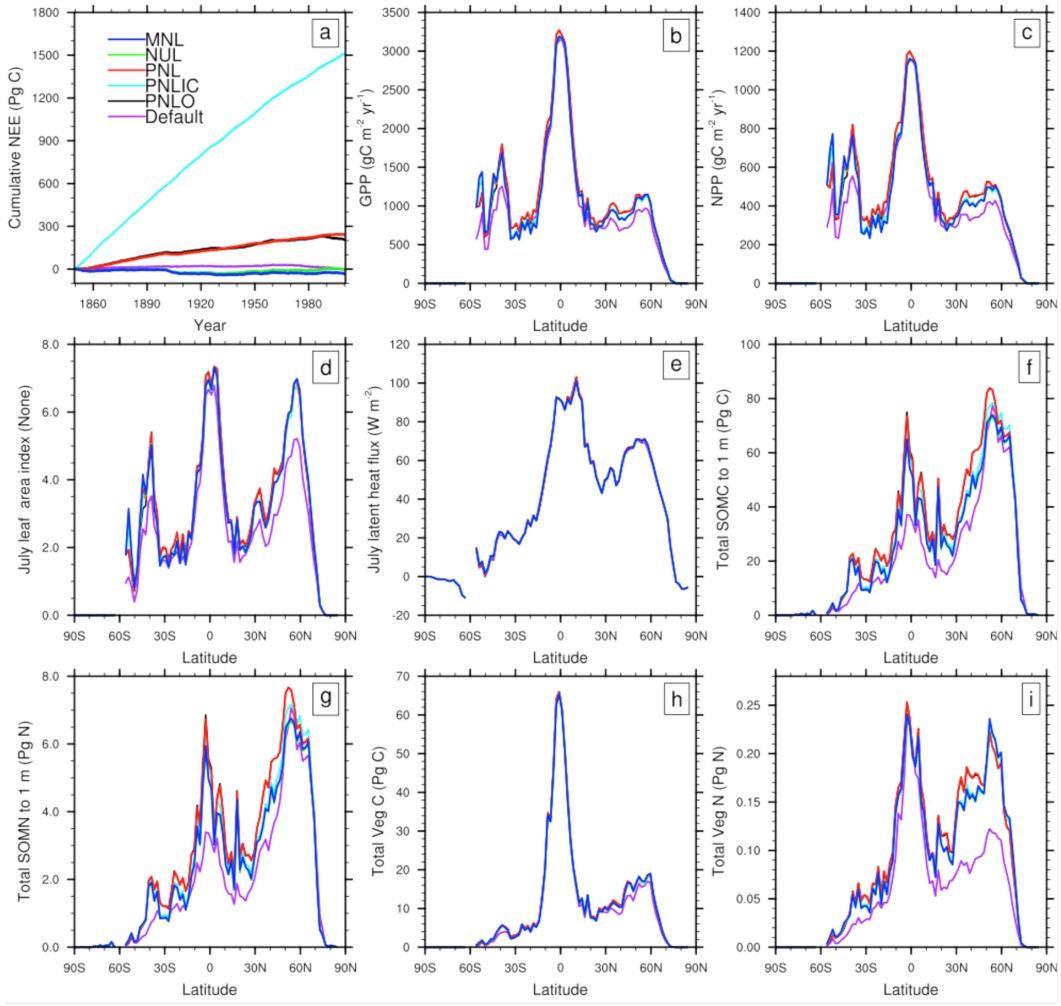

Figure 1. Model predictions for the contemporary period 1850-2000: (a) Cumulative net ecosystem exchange (NEE; positive into the atmosphere); (b) Gross primary productivity; (c) Net primary productivity; (d) July leaf area index; (e) July latent heat flux; (f) total organic soil carbon to 1 m depth; (g) total organic soil nitrogen to 1 m depth; (h) total vegetation carbon; and (i) total vegetation nitrogen. Results for (b)-(i) are averaged over 1991-2000.





Figure 2. Model simulations forced by the Representative Concentration Pathway 4.5 (RCP4.5) atmospheric $CO_2$ for year 2001-2300. Here total soil carbon includes litter carbon and soil organic matter as defined in CLM4.5; coarse woody debris is excluded (Oleson et al., 2013). All changes are calculated as relative to each of their initial carbon pool sizes at the start of the simulation (year 2000).





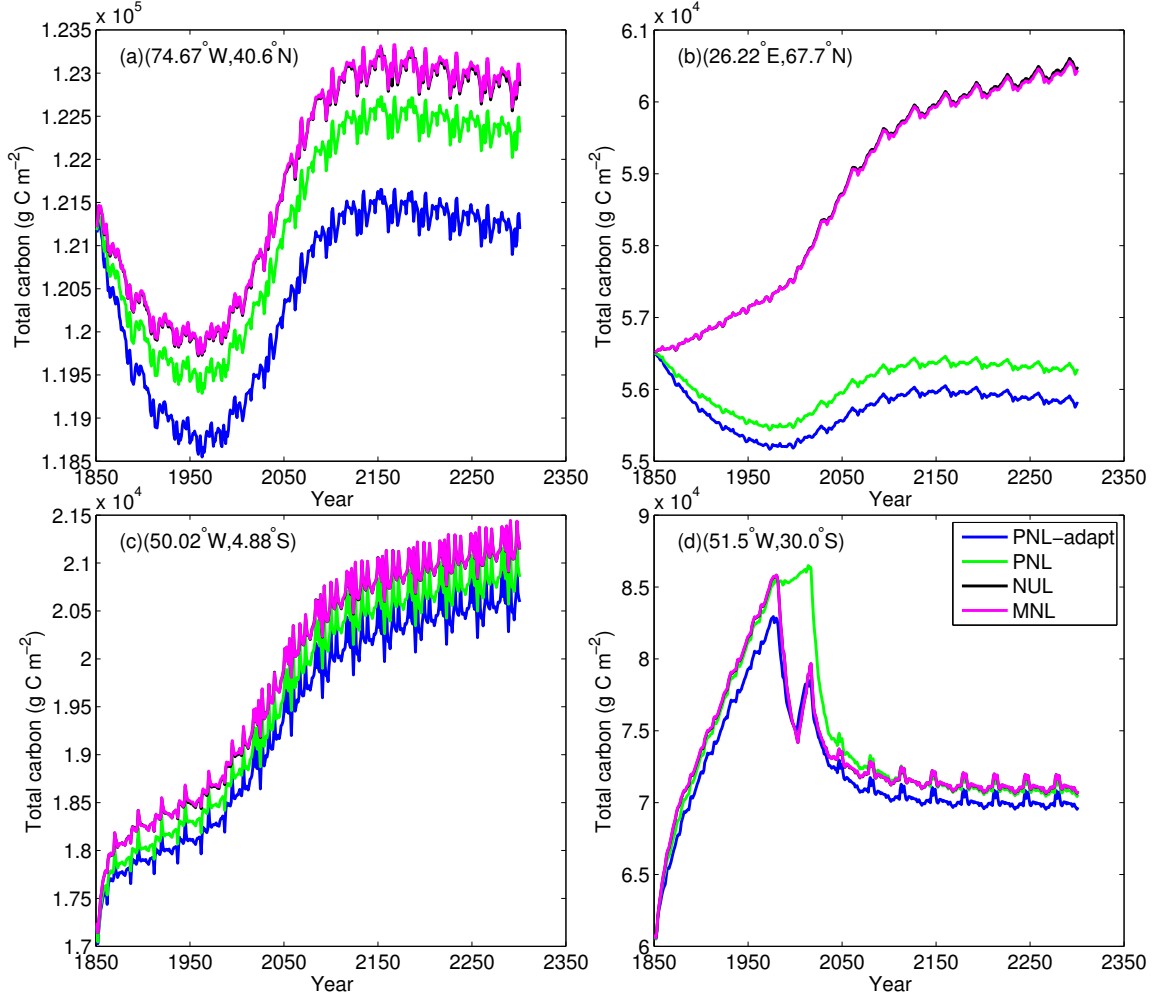

Figure 3. Point simulations for the 4 specific gridcells using different model configratuions. For each site, all simulations used identical initial conditions obtained from spinup with the PNL-adapt code. Note the color schemes are different from that in Figure 1 and Figure 2.



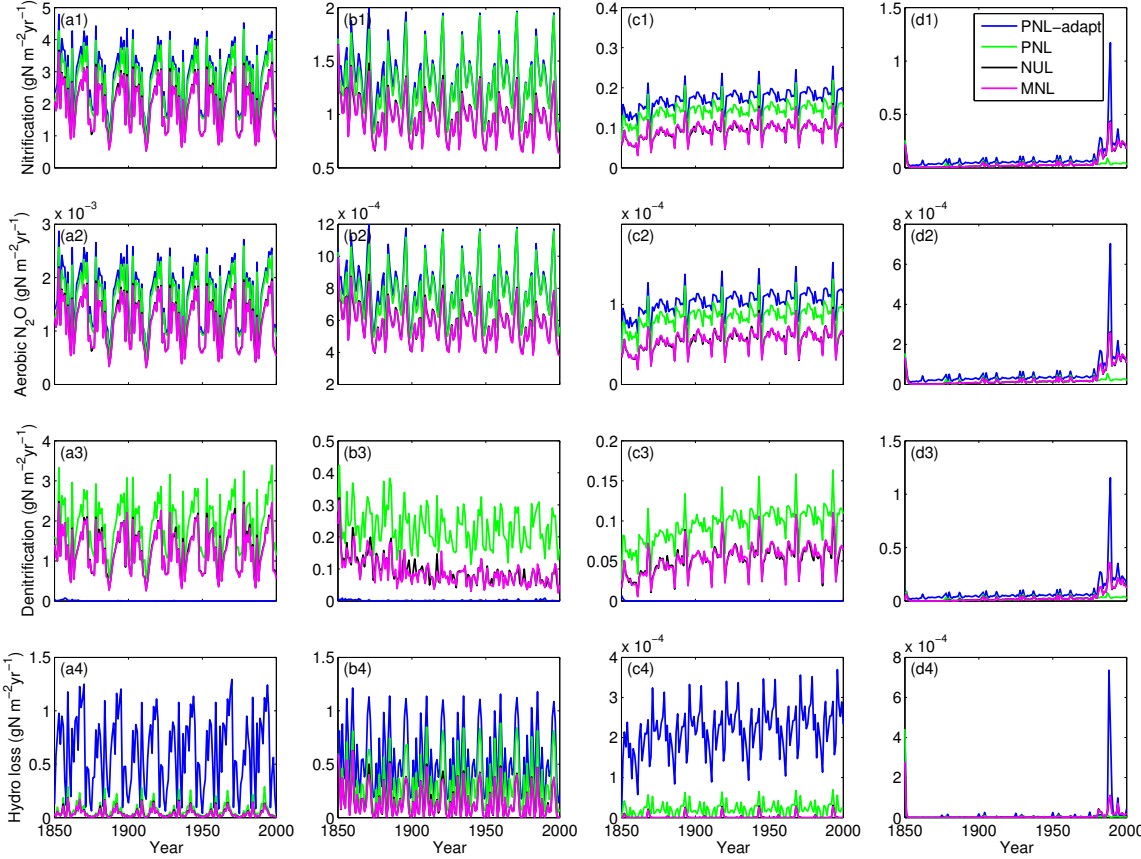

Figure 4. Nitrogen fluxes for the four specific gridcell simulations obtained from different model configurations. The four
columns from left to right correspond to the four locations specified in Figure 3.





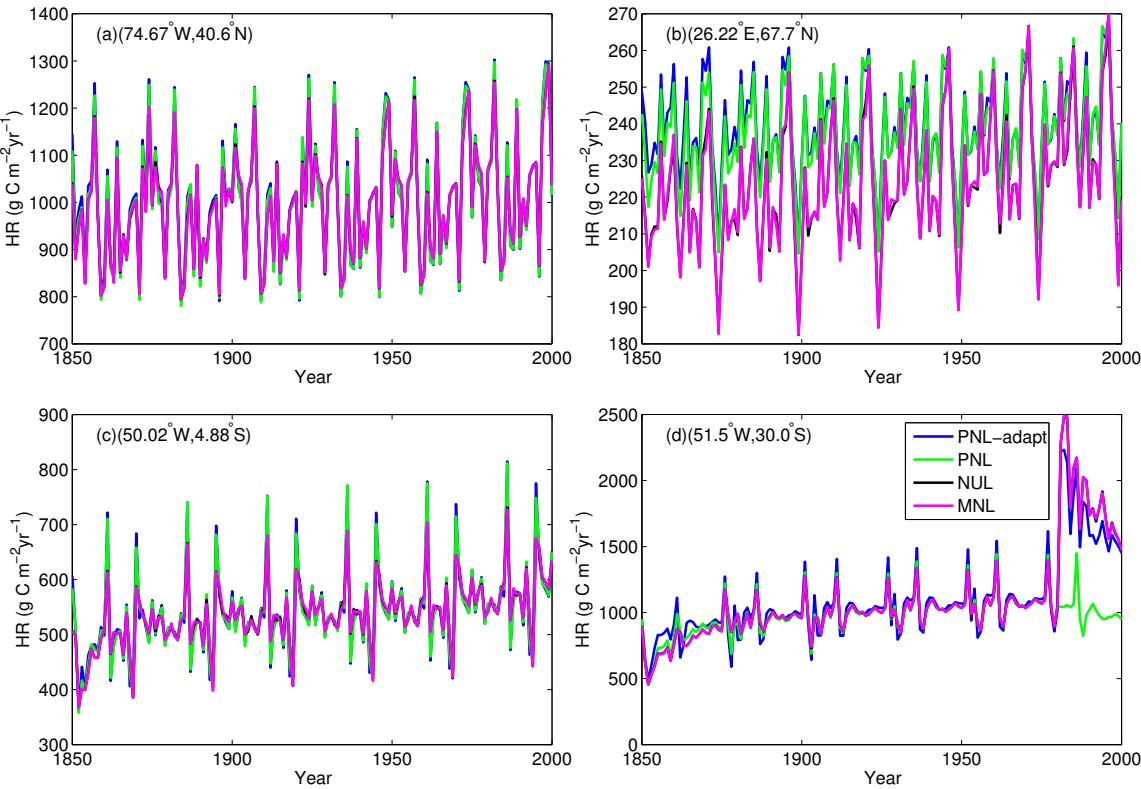

5   Figure 5. Heterotrophic respiraiton for the four specific gridcell simulations obtained from running different model configurations.