# Peer review of "Large uncertainty in ecosystem carbon dynamics resulting from ambiguous numerical coupling of carbon and nitrogen biogeochemistry: A demonstration with the ACME land model"

_Biogeosciences, 2016_

## Referee Comment (RC1) · Anonymous Referee #1 · 13 Jul 2016

I find this study intriguing. There has been a debate about the definition of nutrient limitation (see Davidson and Howarth 2007; Elser et al. 2007 and many more), different representations of nitrogen limitation in numerical models simply reflect those diverging views. What implications of different numerical representations of nitrogen limitation will have on the projected land carbon sink is an important question, and needs a careful study. This study found significant discrepancies in the projected land sinks by ALM using different representations of nitrogen limitation. The results are interesting. However little explanation has been given to why they are different. I also found some results quite intriguing. The other issue identified in this study is double counting of nitrogen limitation. This has been pointed out by others before (Downing et al. 1999; Agren et al. 2012 for example). The issue of double-counting is less prevalent, as several global nutrient models, OCN, CABLE and GFDL land models do not use CLM-like approach, ie reducing GPP when nitrogen demand by plants is higher than available N. OCN and CABLE will vary allocation and tissue chemistry, which will affect GPP, canopy LAI from next time step on. Some preliminary comments are listed as below: 1. The title: Given several caveats of this study, the title is misleading. The "large" uncertainty can result from lack of adequate model calibration, initialization and so on. Even these uncertainty is large for ALM, and may not be for other models. 2. P1, L8. "Abstract" "Most earth system models (ESM). . ." . That is not true, essentially only one model includes N cycle among all AR5 ESMs. 3. P1, L15-16. Comparing the divergence here that is supposed to be caused by different approaches of N limitation with the divergence among mostly carbon-only model is not appropriate. 4. P1, L20-21. "..significant sensitivity of model prediction to initial conditions. . .". For each representation of N limitation, how different are the equilibrium pool sizes and fluxes? If you did not spin each representation to steady state separately, the issue here may be related to initialization and calibration (GPP being too high in this study), not initial values. 5. P2, L1-11. I do not really appreciate the rationale for classifying the "errors" identified in this study into a combination of type I and II. The "errors" simply result from model structure differences. To some extent, errors in numerical implementation can be part of model structure error. I found the identification of four-stages of model design unhelpful. The authors did not follow each of these four stages through in this study, as they did not calibrate the different representations. If they have calibrated different representation using same datasets, the divergence among different representations may be much smaller, and the conclusions from this study may not be accurate any more. Given this caveat, results from this study are better suited for a technical note for ALM model development community. 6. P2, L28-29. Here you stated: "numerical

implementation of a given formulation" is the focus of this study. What are your given formulation? Equations (3), (6), (7) and (8) are mathematically different? I think that your study is about different implementations of nutrient limitation effect, not numerical implementation of the same equations. 7. P3, L14-15. This is not how nutrient limitation is defined in several others global land models. Nutrient limitation can occur even if the nutrient demand is met by uptake. For example, in a fast-growing plantation, the plants will try to increase its LAI first, then its leaf N:C ratio, or both. If LAI increases first, the leaf N:C ratio is low, the canopy photosynthesis is considered to be N limited because adding N fertilizer will increase canopy photosynthesis by increasing leaf N:C ratio, or canopy LAI or both. The CLM-alike approach is not adopted by most other global land models. You should not generalize it to other models here. 8. P3, L25 ".. substrate production is independent of consumption, a situation that occurs exactly in the CENTURY-like models". That is incorrect. If true, progressive nitrogen limitation will not happen in CENTURY-like models, such as G'DAY. 9. P4, L13-15. But S is a function of N mineralization rate as stated in eqn (1). I disagree with your interpretation here. 10. P5. L17. "applying the flux adjustment only to Fs,uptake". By authors' argument, will this also constitute a double–counting of nutrient limitation? 11. P5, Write eqn (7) using notation of t, t+1, or implicit form. 12. P5, eqn (6) and (7), I really not see much differences between these two equations in practice. One can also argue that both N input and available mineral soil N are available for plant uptake in the NUL formulation. 13. P6, Eqn (8). This is an incorrect interpretation of eqn (C12) of Wang et al. (2010). Wang et al. (2010) did not represent N uptake by decomposers explicitly. 14. P6, L18-25. After all, you treated all three approaches as being valid, which contradicts to your earlier arguments that MNL counts for nutrient limitation twice, and NUL requiring flux adjustment that also constitutes double counting of nutrient limitation based on authors' argument. 15. P6. "ambiguous numerical implementation"? Numerical implementation is not ambiguous, but its interpretation is. 16. P7, L23-30. You removed the down-regulation of GPP. That is theoretically better. However you did not re-calibrate your GPP, therefore your estimated plant N demand is excessive,
and may not be met at available soil N. This could be the cause for the oscillatory responses shown in Figure 2. At a given time step, if available soil N plus mineralized N is less than the N demand by plant and microbes, you have to use flexible C:N ratio approach, independent of whichever numerical representations. Here it is important to state whether you have flexible C:N ratios for all pools, and what are the ranges of C:N ratios? What do you do when demand by plants and microbes is higher than available soil mineral N and mineralized N at a given time step? And how different numerical representation deal with this issue while maintaining mass conservation. 17. P7, L25. If you simply remove this down-regulation without tuning your model properly, you will have very high N demand in your model, which likely causes much numerical issues in your integration, such as mass conservation. What you should do is to reduce the potential GPP calculated by your model by calibration. 18. P8. L1-9. CENTURY-like models do not allow any preference by plants or soil microbes between NH4 and NO3. This is not a CENTRY-thing. 19. P8, L10-22. When using each of five different numerical implementations, did you spin the model to steady state for each of them? I do not think that PNLIC is a valid one. 20. P8, L18 ".. finally applied nitrogen limitation to microbes and plants a second time". How? Give more details here. What is the justification of applying nitrogen limitation twice? 21. P9, Section 2.2. How can you use the Qian et al's data of 1848 to 1972 to generate the forcings from 1850 to 2005 for ALM? Here you stated that all model simulations span to steady state at 1850. How different are the steady state pools and fluxes among different numerical representations at 1850? Why diagnostic atmospheric CO2 concentrations (L4)? How different are your diagnostic CO2 concentrations from the observed CO2 concentrations from 1850 to 2000? Did you include land use change in your simulated land carbon dynamics (L8)? 22. P9, Figure 1. I find the results very puzzling. Given that NPP is similar among six different approaches, soil C is also quite similar except that the red curve is generally higher than others across different latitudes. Can the large differences in the simulated NEE be explained by the differences in the simulated heterotrophic respiration among five different approaches? Does each of the five approaches conserve mass of C and

N? We need this evidence to be convinced that the numerical implementation of all five approaches are accurate. I do not see any relevance of showing latent heat flux here. Also the canopy LAI in the tropics and high latitudes (about 60degree North) is unrealistically high (>6). As a result, your N demand is also unrealistically high. 23. I suspect that the divergent results as shown in Figure 1 may be complicated by the lack of mass conservation for some approaches, therefore it is difficult to separate the effect of not conserving mass from different representations of N limitation on the simulated variables. I think that the authors incorrectly attribute all the differences shown in Figure 1 to the representation of N limitation (also see my comment 9). 24. Among the five approaches, I think that PNLIC being invalid and PNLO being a different issue. I suggest that authors remove the results from those two approaches. The presentation of the results, particularly in Figures 1 and 2 are very difficult to distinguish. 25. Figure 1. All six approaches simulated very similar GPP, NPP, soil carbon, but the cumulated NEE by PNLIC is 50 times greater than most other approaches? Where does this huge amount of carbon come from? Please show changes of global carbon pools (vegetation, soil, litter) as well as fluxes in this Figure. Has mass been conserved in all approaches. If not, then the results are not valid. 26. P10. Section 3.2 and Figure 2. Even being averaged over such broad regions (north temperate, tropics and artic), the results still show some periodic oscillation. This needs some detailed explanation. How can we have any confidence in any of the results if masses of C and N are not conserved? Why the changes in vegetation and soil carbon (shown in a2 and a3) do not add up to total carbon change (a1)? Similarly for other two regions as well. 27. I do not know how much of the results are applicable to other models. I think that the authors oversell their results a bit by using very high GPP, therefore high N demand, which differs from other global models. If a more realistic GPP, therefore N demand are used, will the differences among different approaches still be so large? 28. Calibration is another issue. You need to calibrate ALM with each of five approaches properly. If we take any model, and replace part of this model with the formulation from another model, there will be almost infinite number of studies of this kind. The question is how

useful this kind of study really is? 29. The fonts used in the manuscript are hardly readable, quality of several figures are poor (1, 4, 5).

---

## Referee Comment (RC2) · Anonymous Referee #2 · 9 Aug 2016

Review of Tang and Riley

Given the current focus on explaining the large spread in carbon cycle predictions in CMIP5 simulations, studies such as this manuscript help clarify potential drivers of differences. Furthermore, it is important to highlight how subtle differences in process or implementation can potentially lead to large differences in terrestrial carbon stocks. This manuscript focuses on a seemingly subtle difference in how nutrient limitation is executed in a global biogeochemical model. While the authors highlight the issue

as mostly numerical, they are addressing a larger issue in ecosystem modeling that centers on whether plants, microbes, or hydrologic losses have first access to mineral nitrogen in soil solution. I think that the manuscript hides this question in the technical language about substrates, ambiguous coupling, and the equations. This technical detail is important but the paper will likely have a stronger impact if the issue was spelled out as a plant vs. microbe competition. The plant vs. microbe competition issue seems to be the key story of the manuscript, rather than the numerical issues, because the MNL and NUL simulations are very close (i.e., the lines from the simulations cover each other in the figure) with the big difference between the those and the PNL simulations. It seems that the MNL (or NUL) vs. PNL approaches represent two different ecological hypotheses about how the world works and the paper could explore the implications of these plant/microbe competition hypotheses on carbon cycling at the global scale. Such a focus would be easier to follow and provide a clearer and, in my opinion, more valuable contribution to the literature. Overall, the simulations are there but a recasting of the motivation (including reviewing the literature on plant –microbe priority for nitrogen) and an expansion of the discussion is needed.

I do have a concern about the level of detail used to examine the simulations. For example, the comparing Figure 2 suggest that there is missing carbon (total carbon != vegetation + soil) (North temperate MNL: 40 != 8 + 4). Is the missing terrestrial carbon important in the story? It is likely related to the dynamics of the CWD stocks because CWD is accounted for in the total carbon but not in the vegetation or soil carbon. Because of this issue and the (unrealistic?) PNLIC example, more discussion of the CWD dynamics is needed (i.e., how is nutrient limitation of CWD decomposition simulated?). Before the manuscript can be a useful contribution, this missing carbon and CWD issue needs to be explored in detail because it appears to be the primary driver of the differences. Otherwise the differences between the MNL vs PNL simulations at 2300 are small ($\sim$4 Pg C change in north temperate– no change in vegetation + 4 Pg change in soil) and the differences are even smaller at 2100 ($\sim$2.5 Pg C change). Overall, I am left wondering why the MNL/NUL and the PNL simulations are so different and

how it relates to CWD dynamics. In summary, more ecological insight as to why the simulations are different is needed for the manuscript to useful to a broader modeling community.

Additional Comments:

Currently the discussion is not well connected to the results section. The bulk of the discussion is focused on recommendations that do not directly reference or build off particular results of the paper. It causes the manuscript to read like a modeling study that is followed by an opinion paper. I recommend exploring the microbe vs. plant competition issue in more detail and tying the discussion points to specific results.

The introduction sets up two hypotheses without specifying how the hypotheses could be rejected. In a typical ecological study, there is an implied p-value that is used for hypothesis testing. In this simulation study without standard statistics, what is the criteria for accepting or rejecting the hypotheses? I recommend either being more specific with the criteria or shifting away from the hypothesis testing approach and more to addressing questions.

The motivation for using simulations that run to 2300 is not clear. It is hard to put the magnitude of sensitivity in context because the carbon storage out to 2100 is more commonly discussed. How does the spread between the simulations compared to CMIP5 model to model variation at 2100?

From a mass balance approach, the substrate equation 1 is incomplete. Why are losses that are not associated with uptake excluded from the equation?

The PNL simulation in Table 1 states that there is equal competition between plant and microbes. How is the equal competition implemented? (it is not clear from equation 7). Also, does this imply that MNL and NUL have competition that is not equal. I recommend clarifying the assumptions of competition in all the simulations.

Table 1 includes the default simulation but does not highlight how it is different from the

other simulations.

Page 4 Line 11: I recommend the phase 'the to be released. . ..ACME-v1' be removed because it will quickly date the manuscript and who knows if the models will change before the manuscript is published.

Page 7 Line 30: If the down-regulation of GPP was removed, how was vegetation carbon limited by nutrient availability? If the uptake of carbon is not limited by nitrogen but there is not enough N in the soil to grow plant tissue, there will be a build of labile carbon in vegetation and the C:N ratio of vegetation will increase.

Page 9 Line 17: Figure 1a is NEE but NEE is not discussed in the sentence.

Page 11 Line 2: The counter-intuitive result was not discussed in Section 4.1. Please be more explicit in the connections in the discussion

Section 3.4. This section does not add anything to the manuscript and I recommend removing (see discussion above about hypothesis testing)

Figure 3. I recommend using the same colors for each simulation throughout the figures. The colors switch between Figure 2 and 3. (the captions says that the colors changed but it is better for the reader to go ahead and match the colors).

---

## Author Comment (AC1) · 14 Sep 2016

While we have carefully and thoroughly addressed the reviewers comments in the attached pdf file and the revised manuscript with tracked changes, we have the general repose posted below.

Overall response:

While we appreciate the reviewer's time to review our study, we believe the reviewer

has mis-construed significant parts of our arguments and results. We address all the comments below, but note in particular that the reviewer's repeated contention that calibration can make a numerically inconsistent model useful for projecting carbon-climate feedbacks highlights why we think out study is important for the modeling community. We make the point in our paper that, at the most basic level, models require that the numerical encoding is consistent with their analytical formulations. The practice of ensuring this consistency has been standard in other branches of earth system modeling, including atmospheric physics (e.g. Phillips, 1956; Arakawa, 1965; Wan et al., 2016), atmospheric chemistry (Sandu, 2001; Nguyen et al., 2009; Wan et al., 2013), hydrology (Tang et al., 2015) and marine biogeochemistry (Broekhuizen et al., 2008); land biogeochemical modeling should be no exception. Consistent and robust numerical encoding can help ensure that new mechanisms and processes are added for the right reasons, and can remove the false security generated by calibration of structurally uncertain biogeochemical models. Further, our study shows that numerically inconsistent models can result in very misleading predictions of how land ecosystems respond to increasing atmospheric CO2. If the reviewer's opinions on the appropriate use of calibration are widespread in the modeling community (which we believe is the case), we contend that our paper is very relevant and important, in that it dispels those notions and proposes constructive remedies. With the spirit to raise sufficient awareness of these important issues, we carefully address the reviewer's comments point by point in the attached pdf file.

Please also note the supplement to this comment:
http://www.biogeosciences-discuss.net/bg-2016-233/bg-2016-233-AC1-supplement.pdf

**Supplement:**

**Overall response:**

While we appreciate the reviewer's time to review our study, we believe the reviewer has mis-construed significant parts of our arguments and results. We address all the comments below, but note in particular that the reviewer's repeated contention that calibration can make a numerically inconsistent model useful for projecting carbonclimate feedbacks highlights why we think out study is important for the modeling community. We make the point in our paper that, at the most basic level, models require that the numerical encoding is consistent with their analytical formulations. The practice of ensuring this consistency has been standard in other branches of earth system modeling, including atmospheric physics (e.g. Phillips, 1956; Arakawa, 1965; Wan et al., 2016), atmospheric chemistry (Sandu, 2001; Nguyen et al., 2009; Wan et al., 2013), hydrology (Tang et al., 2015) and marine biogeochemistry (Broekhuizen et al., 2008); land biogeochemical modeling should be no exception. Consistent and robust numerical encoding can help ensure that new mechanisms and processes are added for the right reasons, and can remove the false security generated by calibration of structurally uncertain biogeochemical models. Further, our study shows that numerically inconsistent models can result in very misleading predictions of how land ecosystems respond to increasing atmospheric  $CO_2$ . If the reviewer's opinions on the appropriate use of calibration are widespread in the modeling community (which we believe is the case), we contend that our paper is very relevant and important, in that it dispels those notions and proposes constructive remedies. With the spirit to raise sufficient awareness of these important issues, we carefully address the reviewer's comments point by point below.

**Comment 1**: I find this study intriguing. There has been a debate about the definition of nutrient limitation (see Davidson and Howarth 2007; Elser et al. 2007 and many more), different representations of nitrogen limitation in numerical models simply reflect those diverging views. What implications of different numerical representations of nitrogen limitation will have on the projected land carbon sink is an important question, and needs a careful study. This study found significant discrepancies in the projected land sinks by ALM using different representations of nitrogen limitation. The results are interesting. However little explanation has been given to why they are different. I also found some

results quite intriguing. The other issue identified in this study is double counting of nitrogen limitation. This has been pointed out by others before (Downing et al. 1999; Agren et al. 2012 for example). The issue of double-counting is less prevalent, as several global nutrient models, OCN, CABLE and GFDL land models do not use CLM-like approach, i.e. reducing GPP when nitrogen demand by plants is higher than available N. OCN and CABLE will vary allocation and tissue chemistry, which will affect GPP, canopy LAI from next time step on.

**Response:** We thank the reviewer for taking his/her time to review our manuscript and we appreciate his/her positive comments. At the broadest level, there are two aspects associated with how nitrogen limitation is implemented in models: the analytical formulation of the mechanisms and the numerical implementation of those mechanisms. The aspect that the reviewer mentioned above refers to the first (the analytical formulation of nitrogen limitation controls, or growth-controlling in the term recommend by Kovarova-Kovar and Egli (1998)), and the CLM-like formulation is just one of the many formulations used. We focused our study on the second (numerical aspects of the CLM-like implementation), and demonstrated that an inconsistent numerical implementation (Mineral Nitrogen based Limitation (MNL) or Net Uptake based Limitation (NUL)) of the CLM-like formulation resulted in large differences in simulated cumulative land carbon fluxes when compared to that simulated from a consistent numerical implementation (Proportional Nitrogen Limitation (PNL)). Throughout the revised text, we clarify these points.

Second, as for explanations as to how the differences arise, we refer the reviewer to section 4.1 in our first submission and also the revised text, where we applied single-point simulations to investigate these differences. The explanation is quite simple: when both nitrate and ammonium are explicitly competed, the PNL approach resulted in higher nitrification rates (as compared to MNL or NUL), as it allows the newly released ammonia from decomposition to be nitrified, which further enhances nitrogen losses through denitrification and hydrological losses. When this difference is convolved with the high CN ratio of plants, the differences are amplified in the carbon dynamics. In the revised text, we enhance our discussion on how these large differences arise (see section 4.1 and 4.2).

Third, we carefully checked the papers on double counting of nitrogen limitation brought up by the reviewer. We found the "double counting" referred to in these papers is a different concept from the one described in our submitted paper, and that Downing et al. (1999) and Agren et al. (2012) were each taking different viewpoints on different subjects. Downing et al. (1999) were discussing how to experimentally estimate the limitation effect in a phytoplankton fertilization experiment, and concluded that bulk cell biomass rather than bulk growth rate should be used for calculating the nutrient limitation effect. Agren et al. (2012) were analyzing how nutrient co-limitation should be analytically formulated for individual plants. We highlighted these differences in our revised text to ensure other readers will not have this confusion (P6, L7-8).

In contrast, our statement of double counting is that there is a local limitation imposed on each individual through the analytical formulations (using law of the minimum), whereas the ultimate limitation is coming from the interactions of all competitors in the system. So if the law of the minimum is used for modeling nutrient limitation, it should be imposed at the system level, which will result in consistent constraints on individuals in the network. None of the papers referred by the reviewers have pointed out this issue. For this reason, as long as (1) some form of law of the minimum is implemented in a model and (2) the nutrient levels will reach some negative value (if fluxes are not corrected) at some simulation step, double counting will always occur. Given that the law of the minimum is applied to individuals without simultaneous consideration of the overall system constraints in all BGC models we are aware of, this issue is not unique to the CLM-like approach.

**Comment 2:** The title: Given several caveats of this study, the title is misleading. The "large" uncertainty can result from lack of adequate model calibration, initialization and so on. Even this uncertainty is large for ALM, and may not be for other models. **Response**: To make our assertion more conservative, we revised the title as "Potentially large uncertainty in ecosystem carbon dynamics resulting from ambiguous numerical coupling of carbon and nitrogen biogeochemistry: A demonstration with the ACME land model."

**Comment 3**: P1, L8. "Abstract" "Most earth system models (ESM): : :". That is not true, essentially only one model includes N cycle among all AR5 ESMs.

**Response**: Our description is accurate, since, subsequent to the CMIP5 exercise, most models are incorporating the N cycle, and even P cycle, in their development. Some published references are CABEL (Wang et al., 2010) that is coupled to CSIRO Mk3L climate system model, JSBACH (Goll et al., 2012), UVic ESCM (Wania et al., 2012), CESM, and ACME (which we refer to in this study).

**Comment 4**: P1, L15-16. Comparing the divergence here that is supposed to be caused by different approaches of N limitation with the divergence among mostly carbon-only model is not appropriate.

**Respond**: Since we are comparing different uncertainty sources, we believe this comparison is appropriate as has been shown in previous studies (Clein et al., 2007; Huntingford et al., 2013; Tang and Zhuang, 2008). Further, the importance of developing better models of nitrogen dynamics is clear if the divergence from models with nitrogen dynamics is comparable to or larger than that of the carbon only model.

**Comment 5**: P1, L20-21. "...significant sensitivity of model prediction to initial conditions: : :". For each representation of N limitation, how different are the equilibrium pool sizes and fluxes? If you did not spin each representation to steady state separately, the issue here may be related to initialization and calibration (GPP being too high in this study), not initial values.

**Response**: First, we have spun up all simulations separately, and the results indicate that their differences are small (for a few example variables, see Figures X1-X4 at the end of this response, where the spatial distribution of vegetation carbon, vegetation nitrogen, soil carbon and soil nitrogen were found coherent among different models). These small differences are also consistent with the small differences we show in Figure 1 for models excluding PNLIC.

Second, we do not think the calibration issue is relevant here, as we are using the same parameter values of the default model, and the reported variables in the 1990s are

very similar between our new simulations and the default model. We do note that PNLIC is an outlier from the other simulations. This significant difference results from our purposeful running of the PNL model using initial conditions from NUL. In this way, as we explained, PNLIC is used to demonstrate the possible model misuse when the numerical implementation of the model is not well acknowledged to the user. In particular, the rationale behind this numerical experiment is that if we assume two modelers are given the same set of equations to solve, they will obtain very similar results, such that the initial conditions between these two models can be legitimately switched. However, our survey and evaluation in this study (see also Tang and Riley, 2016) indicates that this may not be the case. We also highlight that many modeling papers have not explicitly described their model's numerical details, which may lead to non-reproducible predictions if one tries to recreate the model from scratch and use it for the same type of model simulations.

**Comment 6**: P2, L1-11. I do not really appreciate the rationale for classifying the "errors" identified in this study into a combination of type I and II. The "errors" simply result from model structure differences. To some extent, errors in numerical implementation can be part of model structure error. I found the identification of four-stages of model design unhelpful. The authors did not follow each of these four stages through in this study, as they did not calibrate the different representations. If they have calibrated different representation using same datasets, the divergence among different representations may be much smaller, and the conclusions from this study may not be accurate any more. Given this caveat, results from this study are better suited for a technical note for ALM model development community.

**Response**: While we agree that classifying the "errors" identified in this study into a combination of type I and II may be a personal preference, we do not agree with the loose definition of model structural differences advocated by the reviewer. If considering the "model structure" as any difference in the model's encoding (e.g., numerical methods), then if the model parameterization is hardwired in the model (as many models do), the model parameterization uncertainty can be regarded as model structural differences as well. We therefore, in the spirit of rigorous model development and application, suggest

that the four stages of model design are necessary and helpful, and are more logical than over generalizing the concept of structural uncertainty. In a nutshell, model development and application is one manifestation of applied mathematics, which involves analytical formulation, numerical discretization, parameterization, and application, which are accordant with the four stages we classified. Given this argument, the classification of our identified errors into a combination of type I and II is logical and appropriate.

Further, we think the reviewer has exaggerated the effect and usefulness of model calibration. In all types of modeling work that involve the solution of differential equations, the first step should be to solve the equation in a numerically consistent way. This is why Lax and Richmyer (1956) proposed Lax's equivalence theorem on numerical consistency, which has been the golden standard for solving differential equations (e.g. Smith, 1985). When the model is not solved in a numerically consistent way, calibration is a waste of resources.

The problem we identified here is not unique to ALM (as we argued previously and elaborate further here). On the contrary, the problem of mis-coupling of various modeling components (or physical processes) has been identified in the modeling of atmospheric physics (Wan et al., 2016), atmospheric chemistry (Sanddu 2001; Nguyen et al., 2009; Wan et al., 2013), hydrology (Tang et al., 2015), marine biogeochemistry (Broekhuizen et al., 2008), combustion systems (Gou et al., 2009), and many others. We have added this argument and citations to the revised manuscript. There are even workshops specifically discussing this type of problem (e.g., Workshop on Physics Dynamics Coupling

(http://events.pnnl.gov/default.aspx?topic=Physics\_Dynamics\_Coupling\_in\_Weather\_an d\_Climate\_Models). Therefore, misinterpreting that our results are unique to ALM will further hide such issues in land biogeochemical modeling, rather than to help resolving the large uncertainty in current and maybe future predictions of carbon-climate feedbacks.

Comment 7: P2, L28-29. Here you stated: "numerical implementation of a given formulation" is the focus of this study. What are your given formulation? Equations (3), (6), (7) and (8) are mathematically different? I think that your study is about different

implementations of nutrient limitation effect, not numerical implementation of the same equations.

**Response**: The formulation we are referring to is the land biogeochemical model of CLM4.5 and ALMv0, which is documented very thoroughly in the technical note (Oleson et al., 2013). Since equations (6), (7), and (8) attempt to numerically approximate the same equation, i.e. equation (2), they are therefore different numerical implementations of the same equation.

**Comment 8**: P3, L14-15. This is not how nutrient limitation is defined in several others global land models. Nutrient limitation can occur even if the nutrient demand is met by uptake. For example, in a fast-growing plantation, the plants will try to increase its LAI first, then its leaf N:C ratio, or both. If LAI increases first, the leaf N:C ratio is low, the canopy photosynthesis is considered to be N limited because adding N fertilizer will increase canopy photosynthesis by increasing leaf N:C ratio, or canopy LAI or both. The CLM alike approach is not adopted by most other global land models. You should not generalize it to other models here.

**Response**: We clarified in our original submission (P2, L26-29) that there are two aspects of nutrient limitation: one is the analytical formulation and the other is the numerical implementation. The analytical formulation (e.g., that implemented in CLM4.5) includes all aspects that the reviewer is mentioning here. The numerical implementation specifically refers to the study we describe here. We have revised the paper to further clarify this issue (by adding appropriate references and more descriptions, P2 L29-33, P3:L1-L12). Also, we note a few other models have also adopted the CLM/ALM-like approach, e.g. Biome-BGC (Thornton et al., 2002), BiomeBGC MuSo (Hidy and Barcza, 2014), and JSBACH (Parida, 2011; Goll et al., 2012). So we are not over generalizing our results as inferred by the reviewer.

**Comment 9**: P3, L25 "...substrate production is independent of consumption, a situation that occurs exactly in the CENTURY-like models". That is incorrect. If true, progressive nitrogen limitation will not happen in CENTURY-like models, such as G'DAY. **Response**: We mentioned that there are two aspects of nitrogen limitation: the

formulation aspect is referred throughout the reviewer's comments, whereas we are only discussing the numerical aspects. Therefore, as long as nitrogen dynamics are coupled with carbon dynamics, and nitrogen availability is insufficient to support potential assimilation, progressive nitrogen limitation will always occur. However, what we meant at P3, L25 is that the nitrogen mineralization of some SOM (soil organic matter) pools is not an explicit function of nitrogen uptake from other SOM pools. This dynamic occurs in Century-like models because the different SOM pools decay independently, such that a nitrogen mineralizing pool experiences no nitrogen stress from the concurrently nitrogen immobilizing SOM pools. To avoid this misunderstanding, we add clarifying explanation in our revision (P4: L16-18).

**Comment 10**: P4, L13-15. But S is a function of N mineralization rate as stated in eqn (1). I disagree with your interpretation here.

**Response**: We note that (3) is using the nitrogen pool from the current time step, so our explanation is correct. Also we stated when introducing equation (2) that we are using the forward Euler scheme for model integration. Therefore, this misunderstanding should not arise.

**Comment 11**: P5. L17. "applying the flux adjustment only to Fs,uptake". By authors' argument, will this also constitute a double–counting of nutrient limitation? **Response:** If this flux adjustment is coupled with an explicit use of law of the minimum, then double counting of nutrient limitation can occur. We avoided it in our study by using the scheme described in Tang and Riley (2016).

**Comment 12**: P5, Write eqn (7) using notation of t, t+1, or implicit form. **Response**: We decide to keep it as is in order to avoid further complication and confusion because we stated explicitly that we are using the forward Euler scheme when introducing equation (2).

**Comment 13**: P5, eqn (6) and (7), I really not see much differences between these two equations in practice. One can also argue that both N input and available mineral soil N

are available for plant uptake in the NUL formulation.

**Response**: The difference occurs in the numerical implementation when the limitation strength predicted by equation (6) is stronger than that by equation (7) under the same conditions. We have provided our mathematical proof of this situation in the originally submitted (and also the revised) supplemental material.

Comment 14: P6, Eqn (8). This is an incorrect interpretation of eqn (C12) of Wang et al. (2010). Wang et al. (2010) did not represent N uptake by decomposers explicitly.
Response: We did not understand the reviewer's meaning here. Nevertheless, in Wang et al.'s equation (C12), the model imposes nitrogen limitation based on net unlimited N mineralization, which is used exactly in equation (8) of our submitted paper.

**Comment 15**: P6, L18-25. After all, you treated all three approaches as being valid, which contradicts to your earlier arguments that MNL counts for nutrient limitation twice, and NUL requiring flux adjustment that also constitutes double counting of nutrient limitation based on authors' argument.

**Response**: We believe this is another misunderstanding of our study. The double counting has no direct connection to MNL, NUL, or PNL. Rather the double counting results from first applying law of the minimum on individual consumers, and then rectifying the nutrient fluxes a second time if the nutrient stock would become negative without such rectification. See our response to comment 1 for further discussions.

**Comment 16**: P6. "ambiguous numerical implementation"? Numerical implementation is not ambiguous, but its interpretation is.

**Response**: We acknowledge that numerical implementation may also be a mathematical manifestation of ambiguous interpretation; when the interpretation is ambiguous, so is the numerical implementation, and vice versa.

**Comment 17**: P7, L23-30. You removed the down-regulation of GPP. That is theoretically better. However you did not re-calibrate your GPP, therefore your estimated plant N demand is excessive, and may not be met at available soil N. This could be the

cause for the oscillatory responses shown in Figure 2. At a given time step, if available soil N plus mineralized N is less than the N demand by plant and microbes, you have to use flexible C:N ratio approach, independent of whichever numerical representations. Here it is important to state whether you have flexible C:N ratios for all pools, and what are the ranges of C:N ratios? What do you do when demand by plants and microbes is higher than available soil mineral N and mineralized N at a given time step? And how different numerical representation deal with this issue while maintaining mass conservation.

**Response**: First, the oscillatory response in Figure 2 resulted from the cycling of climate forcing data (we clarified this in the revision by stating it clearly in the figure captions), and has nothing to do with the removal of GPP down-regulation. In the revised text, we state clearly that the version of the model used here applies fixed C:N ratios and that testing with flexible C:N is underway, and will be reported elsewhere. Further, in the models we are comparing, all use the same model parameters, as they are arguably solving the same set of model equations. Therefore, the calibration is not a relevant issue here. Finally, we did check that the revised models are behaving similarly as the default model in the historical period (also see response to comment 5), and have added description to this effect in the revised manuscript.

**Comment 18**: P7, L25. If you simply remove this down-regulation without tuning your model properly, you will have very high N demand in your model, which likely causes much numerical issues in your integration, such as mass conservation. What you should do is to reduce the potential GPP calculated by your model by calibration. **Response**: As we argued previously, the calibration is not a relevant issue in this study,

given that our model results for the contemporary period are quite close to the default CLM4.5 and ALMv0, which have been reviewed, publically released, and applied in many studies. Also, the mass conservation issue raised by the reviewer has been very carefully addressed, and in fact, if the masses of carbon, nitrogen, and water are not conserved, our model will stop and issue an error message to that effect.

**Comment 19**: P8. L1-9. CENTURY-like models do not allow any preference by plants or soil microbes between NH4 and NO3. This is not a CENTRY-thing.

**Response**: We revised the language to make the statement less specific to CENTURYlike models.

**Comment 20**: P8, L10-22. When using each of five different numerical implementations, did you spin the model to steady state for each of them? I do not think that PNLIC is a valid one.

**Response**: Yes, we did careful spinup for all simulations. PNLIC is an intentional simulation with PNL using initial conditions from NUL, which reflects the incompatibility of the two models. This simulation is one example to demonstrate the potential danger if the numerical implementation is not acknowledged to the user. See our response to comment 5 for further discussions.

**Comment 21**: P8, L18 ".. finally applied nitrogen limitation to microbes and plants a second time". How? Give more details here. What is the justification of applying nitrogen limitation twice?

**Response**: We explained in the text that because this is the only way to prevent the model from crashing (through negative nitrogen or oxygen concentrations). This approach is equivalent to the projector-corrector methods, where the first step is making a prediction, and the second step is a correction to impose the actual nitrogen limitation through linear downregulation.

**Comment 22**: P9, Section 2.2. How can you use the Qian et al's data of 1848 to 1972 to generate the forcings from 1850 to 2005 for ALM? Here you stated that all model simulations span to steady state at 1850. How different are the steady state pools and fluxes among different numerical representations at 1850? Why diagnostic atmospheric CO2 concentrations (L4)? How different are your diagnostic CO2 concentrations from the observed CO2 concentrations from 1850 to 2000? Did you include land use change in your simulated land carbon dynamics (L8)?

**Response**: Yes we spun up all models. The diagnostic CO2 concentrations means using

observed atmospheric  $CO_2$  concentrations. We included land use change using the standard approach in CLM4.5, and since all simulations were run with the same protocol, the details of land use change are not relevant here. In addition, this practice is standard in applying CLM or ALM.

**Comment 23**: P9, Figure 1. I find the results very puzzling. Given that NPP is similar among six different approaches, soil C is also quite similar except that the red curve is generally higher than others across different latitudes. Can the large differences in the simulated NEE be explained by the differences in the simulated heterotrophic respiration among five different approaches? Does each of the five approaches conserve mass of C and N? We need this evidence to be convinced that the numerical implementation of all five approaches are accurate. I do not see any relevance of showing latent heat flux here. Also the canopy LAI in the tropics and high latitudes (about 60degree North) is unrealistically high (>6). As a result, your N demand is also unrealistically high. Response: By design, our models maintain rigorous mass balance for carbon, nitrogen, and water. The differences between the models (excluding PNLIC) are within the range of uncertainties as reported in other studies. We are aware that CLM does not simulate reliable LAI because its poor representations of the carbon and nutrient allometry and stoichiometry. These issues are under improvement and results will be reported elsewhere. However, as we explained above and in the revised manuscript, these issues will not change the conclusion of our study.

**Comment 24**: I suspect that the divergent results as shown in Figure 1 may be complicated by the lack of mass conservation for some approaches, therefore it is difficult to separate the effect of not conserving mass from different representations of N limitation on the simulated variables. I think that the authors incorrectly attribute all the differences shown in Figure 1 to the representation of N limitation (also see my comment 9).

**Response**: As we explained, our models have rigorous mass balance checks for carbon, nitrogen, and water. Otherwise, the model will stop and no simulations could be done.

**Comment 25**: Among the five approaches, I think that PNLIC being invalid and PNLO being a different issue. I suggest that authors remove the results from those two approaches. The presentation of the results, particularly in Figures 1 and 2 are very difficult to distinguish.

**Response**: With all our explanations above, we decide to keep PNLIC and PNLO in our results. Also, the other reviewer has no complaints about Figures 1 and 2.

**Comment 26**: Figure 1. All six approaches simulated very similar GPP, NPP, soil carbon, but the cumulated NEE by PNLIC is 50 times greater than most other approaches? Where does this huge amount of carbon come from? Please show changes of global carbon pools (vegetation, soil, litter) as well as fluxes in this Figure. Has mass been conserved in all approaches. If not, then the results are not valid. **Response**: Once again, we addressed the reviewer's concern regarding mass balance in previous comments. The difference is from shifted partitioning between nitrogen losses

and nitrogen uptake: when combined with the usual high CN ratio of vegetation and SOM pools, the small differences in nitrogen turns into large differences in carbon. We have provided some of this extra information in the supplemental material in our first submission, and we enhanced relevant explanations in both the revised text and supplemental material.

**Comment 27**: P10. Section 3.2 and Figure 2. Even being averaged over such broad regions (north temperate, tropics and artic), the results still show some periodic oscillation. This needs some detailed explanation. How can we have any confidence in any of the results if masses of C and N are not conserved? Why the changes in vegetation and soil carbon (shown in a2 and a3) do not add up to total carbon change (a1)? Similarly for other two regions as well.

**Response**: As described above, the periodic oscillation results from the recycling of climate forcing. We further clarified this in our revised manuscript. Once again, the reviewer's questioning on mass balance has been addressed: the models all conserve mass.

**Comment 28**: I do not know how much of the results are applicable to other models. I think that the authors oversell their results a bit by using very high GPP, therefore high N demand, which differs from other global models. If a more realistic GPP, therefore N demand are used, will the differences among different approaches still be so large? **Response**: To the contrary, we believe our results are highly valuable because they are made on the generic point of ambiguous numerical implementation of nutrient effects on the carbon cycle. Our study attempts to bring these issues in land modeling to the forefront, and are analogous with ongoing improvements in other components of earth system models (e.g., Wan et al., 2016)

**Comment 29**: Calibration is another issue. You need to calibrate ALM with each of five approaches properly. If we take any model, and replace part of this model with the formulation from another model, there will be almost infinite number of studies of this kind. The question is how useful this kind of study really is?

**Response**: As we explained previously, calibration is irrelevant in this study, given the similar model behavior in the historical period when comparing our models and the default CLM4.5/ALMv0. And we stress again that it is not a good approach to calibrate an inappropriately implemented model to make it better match observations. We have added text to the revised manuscript to clarify this issue (P3: L11-12).

**Comment 30**: The fonts used in the manuscript are hardly readable, quality of several figures are poor (1, 4, 5).

**Response**: We followed the EGU Copernicous template in preparing our manuscript, and the font size was set as small. We revised Figures 4 and 5 for color consistency. Otherwise, we think Figure 1 is sufficiently clear to serve its purpose, and the other reviewer can read it clearly.

**References**

Agren, G. I., Wetterstedt, J. A. M., and Billberger, M. F. K.: Nutrient limitation on terrestrial plant growth - modeling the interaction between nitrogen and phosphorus, New Phytol, 194, 953-960, 2012.

Arakawa A., Computational design for long-term numerical integration of the equations of fluid motion: Two-dimensional incompressible flow, part I, JCP, 1966. Broekhuizen et al., An improved and generalized second order, unconditionally positive,

mass-conserving integration scheme for biochemical systems, ANM, 2008.

Clein, J., McGuire, A. D., Euskirchen, E. S., and Calef, M.: The effects of different climate input datasets on simulated carbon dynamics in the Western Arctic, Earth Interact, 11, 2007.

Downing, J. A., Osenberg, C. W., and Sarnelle, O.: Meta-analysis of marine nutrientenrichment experiments: Variation in the magnitude of nutrient limitation, Ecology, 80, 1157-1167, 1999.

Gou et al., An efficient multi time scale method for solving stiff ODEs with detailed kinetic mechanisms and multi scale physical chemical processes, AIAA, 2009. Hidy, D. and Barcza, Z.: User's Guide for Biome-BGC MuSo v3.0, Manuscript - revision: 9 September 2014, pp. 43, 2014.

Huntingford, C., Zelazowski, P., Galbraith, D., Mercado, L. M., Sitch, S., Fisher, R.,
Lomas, M., Walker, A. P., Jones, C. D., Booth, B. B. B., Malhi, Y., Hemming, D., Kay,
G., Good, P., Lewis, S. L., Phillips, O. L., Atkin, O. K., Lloyd, J., Gloor, E., ZaragozaCastells, J., Meir, P., Betts, R., Harris, P. P., Nobre, C., Marengo, J., and Cox, P. M.:
Simulated resilience of tropical rainforests to CO2-induced climate change, Nat Geosci,
6, 268-273, 2013.

Kovarova-Kovar, K. and Egli, T.: Growth kinetics of suspended microbial cells: From single-substrate-controlled growth to mixed-substrate kinetics, Microbiol Mol Biol R, 62, 646-666, 1998.

Nguyen et al., Mass conservative, positive definite integrator for atmospheric chemical dynamics, Atmos. Env., 2009.

Oleson et al., Technical description of version 4.5 of the Community Land Model, doi:10.5065/D6RR1W7M, 2013.

Parida, B.: The influence of plant nitrogen availability on the global carbon cycle and N2O emissions, Reports on Earth System Science,

http://www.mpimet.mpg.de/fileadmin/publikationen/Reports/WEB BzE 92.pdf, 2011. Phillips N. A., The general circulation of the atmosphere: a numerical experiment, **QJRMS**, 1956

Sandu A., Positive numerical integration methods for chemical kinetic systems, JCP, 2001.

Smith, G. D., Numerical Solution of Partial Differential Equations: Finite Difference Methods, 3rd ed., Oxford University Press, pp. 67–68, 1985.

Tang, J. Y. and Zhuang, Q. L.: Equifinality in parameterization of process-based biogeochemistry models: A significant uncertainty source to the estimation of regional carbon dynamics, J Geophys Res-Biogeo, 113, 2008.

Tang et al., Incorporating root hydraulic redistribution in CLM4.5: effects on predicted site and global evapotranspiration, soil moisture and water storage, JAMES, 2015.

Tang and Riley, A generic law-of-the-minimum flux limiter for simulating substrate limitation in biogeochemical models, BG, 2016.

Wang, Y. P., Law, R. M., and Pak, B.: A global model of carbon, nitrogen and phosphorus cycles for the terrestrial biosphere, Biogeosciences, 7, 2261-2282, 2010.

Wan et al., Numerical issues associated with compensating and competing processes in climate models: an example from ECHAM-HAM, GMD, 2013.

Wan et al., A new and inexpensive non-bit-for-bit solution reproducibility test based on time step convergence (TSC1.0), GMDD, 2016.

Wania, R., Meissner, K. J., Eby, M., Arora, V. K., Ross, I., and Weaver, A. J.: Carbonnitrogen feedbacks in the UVic ESCM, Geosci Model Dev, 5, 1137-1160, 2012.

---

## Author Comment (AC2) · 14 Sep 2016

Jinyun Tang and William J. Riley

jinyuntang@gmail.com

Comment: Given the current focus on explaining the large spread in carbon cycle predictions in CMIP5 simulations, studies such as this manuscript help clarify potential drivers of differences.

Furthermore, it is important to highlight how subtle differences in process or implementation can potentially lead to large differences in terrestrial carbon stocks. This

manuscript focuses on a seemingly subtle difference in how nutrient limitation is executed in a global biogeochemical model. While the authors highlight the issue as mostly numerical, they are addressing a larger issue in ecosystem modeling that centers on whether plants, microbes, or hydrologic losses have first access to mineral nitrogen in soil solution. I think that the manuscript hides this question in the technical language about substrates, ambiguous coupling, and the equations. This technical detail is important but the paper will likely have a stronger impact if the issue was spelled out as a plant vs. microbe competition. The plant vs. microbe competition issue seems to be the key story of the manuscript, rather than the numerical issues, because the MNL and NUL simulations are very close (i.e., the lines from the simulations cover each other in the figure) with the big difference between those and the PNL simulations. It seems that the MNL (or NUL) vs. PNL approaches represent two different ecological hypotheses about how the world works and the paper could explore the implications of these plant/microbe competition hypotheses on carbon cycling at the global scale. Such a focus would be easier to follow and provide a clearer and, in my opinion, more valuable contribution to the literature. Overall, the simulations are there but a recasting of the motivation (including reviewing the literature on plant –microbe priority for nitrogen) and an expansion of the discussion is needed.

Response: We sincerely appreciate the reviewer's positive comments. We admit that we have hidden the conceptual (or formulation of the growth-controlling) part of the big topic regarding plant vs. microbe competition behind the technical discussion, but we did allude to it: "there are two aspects that determine the modelled influence of nitrogen on ecosystem carbon dynamics". While formulating the growth-controlling mechanisms is important, we believe the numerical part deserves an equal attention, and, more importantly, the numerical part is rarely discussed in land biogeochemical modeling. This is in strong contrast with other branches of earth system modeling, e.g. atmospheric chemistry (Sandu, 200; Wan et al., 2013), atmospheric physics (Arakawa, 1966), hydrology (Tang et al., 2015) and marine biogeochemistry (Broekhuizen et al., 2008). These studies suggest that if the numerical implementation is not consistently

done with respect to the model formulation, what the code conveys is not what the model equations attempt to describe, and when the numerically inconsistent solution is compared to the numerically consistent solution, the difference (manifested as uncertainty) could be surprisingly significant. Like all the examples we are aware of, this significant difference occurred with the common formulation of plant-microbial nitrogen competition (aka the demand-supply ansatz) as used in current and previous versions of CLM, BIOMBGC (Hidy and Barcza, 2014), and JSBACH-CN (Parida, 2011). For the conceptualization or analytical formulation of plant vs microbe nutrient competition, we are well aware that there is currently no consensus about how it should be formulated in biogeochemical models, and this deserves a dedicated analysis integrating both measurements and model formulations. We therefore discussed the conceptual part in detail in another paper we submitted to Ecological Applications. There using detailed measurements in an alpine meadow ecosystem we examined why many of the existing conceptualizations give biased predictions of plant-microbe competition of nutrients and how the plant vs microbe nutrient competition could be improved with a more mechanistic formulation. Nevertheless, per your suggestion, in the revised paper, we covered more references on the conceptual part and gave the readers more contexts upon why we focused on the numeric aspect here (see P2:L 27-34, P3:L1-L12).

Comments: I do have a concern about the level of detail used to examine the simulations. For example, the comparing Figure 2 suggest that there is missing carbon (total carbon != vegetation + soil) (North temperate MNL: 40 != 8 + 4). Is the missing terrestrial carbon important in the story? It is likely related to the dynamics of the CWD stocks because CWD is accounted for in the total carbon but not in the vegetation or soil carbon. Because of this issue and the (unrealistic?) PNLIC example, more discussion of the CWD dynamics is needed (i.e., how is nutrient limitation of CWD decomposition simulated?). Before the manuscript can be a useful contribution, this missing carbon and CWD issue needs to be explored in detail because it appears to be the primary driver of the differences. Otherwise the differences between the MNL vs PNL simulations at 2300 are small (∼4 Pg C change in north temperate– no change

in vegetation + 4 Pg change in soil) and the differences are even smaller at 2100 (∼2.5 Pg C change). Overall, I am left wondering why the MNL/NUL and the PNL simulations are so different and how it relates to CWD dynamics. In summary, more ecological insight as to why the simulations are different is needed for the manuscript to useful to a broader modeling community.

Response: We apologize that we did not present the more detailed exploration of CWD dynamics in the first submission, even though we mentioned it briefly for the historical period in the supplemental material (Figure S3). In ALM, CWD is accumulated from mortality due to fire and background death, harvest and land use change, whereas it is lost through decomposition into lignin and cellulose, a process that is assumed to involve no $CO_2$ release but usually requires nitrogen to proceed. Therefore, the increased decomposition to nitrogen will enhance the loss of CWD, and therefore the overall heterotrophic respiration. For the historical period, we observed that PNLIC has cumulative heterotrophic respiration about 1400 Pg C higher than MNL and NUL, whereas the NPP of PNLIC differs only about 30 Pg C from that by MNL and NUL, therefore, about 700 Pg C of the 1500 Pg C carbon loss from PNLIC is due to enhanced heterotrophic respiration. The other 800 Pg C loss is due to fire, harvest and land use change. For PNL, the cumulative NPP was about 900 Pg C higher than that from MNL and NUL by year 2000. However, the cumulative heterotrophic respiration is about 1400 Pg C higher than that from MNL and NUL by year 2000. This leads to a loss of about 500 Pg C in cumulative NEP, which when integrated with the storage in harvested products leads to an increase of more than 200 Pg C in cumulative emission by 2000 when compared to MNL and NUL. Although the magnitudes are different for the period from 2001-2300, the results indicate CWD is likely an important component to improve in modeling carbon-nutrient feedbacks. Also, in the revision, we noticed a bug in the index function for summarizing regional statistics using NCL. We therefore redraw the time series for different regions using MATLAB. Although the exact number appeared different, the conclusion of our first submission maintained.

Additional Comments:

Comment: Currently the discussion is not well connected to the results section. The bulk of the discussion is focused on recommendations that do not directly reference or build off particular results of the paper. It causes the manuscript to read like a modeling study that is followed by an opinion paper. I recommend exploring the microbe vs. plant competition issue in more detail and tying the discussion points to specific results.

Response: We adjusted the discussion to make it more related to our findings. Specifically, we added section 4.2 for CWD dynamics. However, we decide to keep an appropriate amount of the original discussion in the hopes that we could help other modelers to resolve similar problem in a more systematic way.

Comment: The introduction sets up two hypotheses without specifying how the hypotheses could be rejected. In a typical ecological study, there is an implied p-value that is used for hypothesis testing. In this simulation study without standard statistics, what is the criterion for accepting or rejecting the hypotheses? I recommend either being more specific with the criteria or shifting away from the hypothesis testing approach and more to addressing questions.

Response: We added more contexts on how the hypotheses are tested in the methods (P8: L1-6) and results sections (section 3.4).

Comment: The motivation for using simulations that run to 2300 is not clear. It is hard to put the magnitude of sensitivity in context because the carbon storage out to 2100 is more commonly discussed. How does the spread between the simulations compared to CMIP5 model to model variation at 2100?

Response: Simply put, we want to evaluate whether the models behave drastically differently for very long simulations as compared to short-term simulation. The decision is made in the spirit of pushing the model to extremes. Such a rationale is analogous to field experiments that adding an unrealistic dose of nitrogen fertilizer in one shot,

and see how the system responds. We did compare the spread at 2100 to CMIP5 models analyzed in Shao et al. (2013) and, found they are of similar magnitude (see P10, L21-24 in the first submission).

Comment: From a mass balance approach, the substrate equation 1 is incomplete. Why are losses that are not associated with uptake excluded from the equation? The PNL simulation in Table 1 states that there is equal competition between plant and microbes. How is the equal competition implemented? (it is not clear from equation 7). Also, does this imply that MNL and NUL have competition that is not equal. I recommend clarifying the assumptions of competition in all the simulations.

Response: Please see P3, L9-12 in first submission for a description of these issues. Following Tang et al. (2013), we solved the transport as a separate processes, and equation (1) only represents the competing processes during decomposition. In this study, all schemes assume equal competition between plants and microbes (which is however unlikely to be true as we will present elsewhere). We also clarified these in the revised text (P3: L14-29).

Comment: Table 1 includes the default simulation but does not highlight how it is different from the other simulations.

Response: We included more information on the revised supplemental material and revised text. Briefly, the default simulation is designed to check that the new model implementations are ballpark reasonable.

Comment: Page 4 Line 11: I recommend the phase 'the to be released: : :.ACME-v1' be removed because it will quickly date the manuscript and who knows if the models will change before the manuscript is published.

Response: We revised the text per your suggestion.

Comment: Page 7 Line 30: If the down-regulation of GPP was removed, how was vegetation carbon limited by nutrient availability? If the uptake of carbon is not limited

by nitrogen but there is not enough N in the soil to grow plant tissue, there will be a build of labile carbon in vegetation and the C:N ratio of vegetation will increase.

Response: We clarified that in the revised the scheme GPP is limited the by nitrogen from storage and translocation, so a down-regulation is still occurs but is not as abrupt as the original down-regulation scheme, which reduces GPP instantaneously after accounting for soil nitrogen availability (P8: L30-31).

Comment: Page 9 Line 17: Figure 1a is NEE but NEE is not discussed in the sentence.

Response: We revised the text by removing the reference to Figure 1a.

Comment: Page 11 Line 2: The counter-intuitive result was not discussed in Section 4.1. Please be more explicit in the connections in the discussion

Response: We now made this discussion more explicit.

Comment: Section 3.4. This section does not add anything to the manuscript and I recommend removing (see discussion above about hypothesis testing)

Response: We enriched the revised text so that this section is more meaningful and relevant. Overall, we want to give the readers a clear place to find the conclusion of the hypothesis evaluation.

Comment: Figure 3. I recommend using the same colors for each simulation throughout the figures. The colors switch between Figure 2 and 3. (the captions says that the colors changed but it is better for the reader to go ahead and match the colors).

Response: We adjusted the color per your suggestion.

———————————————————

[Figure]

**Supplement:**

[revised manuscript text omitted]

We answer the above question by focusing on nitrogen—the most important macronutrient related to whether or not terrestrial ecosystems could continue to sequester anthropogenic $CO_2$ (Oren et al., 2001; Drake et al., 2011; Grant, 2013). Following Kovarova-Kova and Egqli (1998)'s use of terms "growth-controlling" and "growth-limiting" in substrate dynamics, we note there are two aspects that determine the modelled influence of nitrogen on ecosystem carbon dynamics: (1) the mechanistic formulation of carbon and nitrogen coupling (i.e. growth-controlling) and (2) the numerical implementation of a given formulation (i.e. growth-limiting). The first aspect regards the analytical formulation of how one or more nutrients mechanistically limit the growth of an organism or a compartment of an organism. We acknowledge that

there has been no consensus on how this should be done, and many opinions are currently under debate (e.g., Kooijman, 1998; Parida, 2011; Agren et al., 2012; Niu et al., 2016; Stocker et al., 2016; Zhu et al., 2016). Therefore, we will present our opinion on this first aspect elsewhere, although some of that has been alluded in Tang and Riley (2013), Zhu and Riley (2015) and Zhu et al. (2016). The second aspect concerns how the coupling between different components as given in an analytical formulation should be achieved in a numerically consistent manner. This second aspect has been rarely discussed in the field of land biogeochemical modelling; event though similar issues (called as multi-physics coupling) have been scrutinized in other branches of earth system modelling. A few excellent examples are Phillips (1956), Arakawa (1965) and Wan et al. (2016) for atmospheric physics, Sandu (2001), Nguyen et al. (2009) and Wan et al. (2013) for atmospheric chemistry, Tang et al. (2015) for soil-plant hydrology and Broekhuizen et al. (2008) for marine biogeochemistry. In a nutshell, all these studies indicate that an inappropriate numerical implementation could render an analytically well-formulated model to behave unrealistically, and calibrating and applying such models (in terms of doing steps III and IV as identified above) will be a waste of resources (as implied in the Lax equivalence theorem (Lax and Richtmyer, 1956)).

For this study, we begin our analysis with the following equation for a generic substrate $S$ in a soil control volume:

$$\frac{dS}{dt} = F_{S,input} - F_{S,uptake} \tag{1}$$

where $F_{S,input}$ and $F_{S,uptake}$ are, respectively, substrate input (from all sources) and substrate uptake (by all competing entities). Here and below, unless otherwise stated explicitly, we assume the units of all variables in a given equation are consistently defined. To simplify the discussion, we have solved the overall spatiotemporal evolution of substrate $S$ (which is a function of both transport and biogeochemistry) using the operator splitting approach (e.g., Strang, 1968; Tang et al., 2013), so that $F_{S,input}$ and $F_{S,uptake}$ in equation (1) only refer, respectively, to substrate release and uptake from the interacting agents. As such, for the substrate $S$ (i.e. mineral nitrogen) that we are interested in (note we henceforth use $S$ and mineral nitrogen interchangeably unless a clarification is required), input $F_{S,input}$ is microbial nitrogen mineralization from soil organic matter (SOM) decomposition; while $F_{S,uptake}$ includes plant nitrogen assimilation (to support growth) and microbial nitrogen utilization (to support decomposition, nitrification and denitrification). If the interaction between soil mineral surfaces and ammonium nitrogen is considered (e.g. Gerber et al., 2010), $F_{S,input}$ and $F_{S,uptake}$ should be modified accordingly, depending on whether ammonium is adsorbed or desorbed from soil minerals. With the operator splitting approach, nitrogen input from other sources (including fertilization, atmospheric nitrogen deposition, nitrogen fixation) and losses through hydrological transport are integrated separately from the competitive coupling between nitrogen mineralization and assimilation. We have tested this treatment by switching the order between solving the biogeochemical processes and transport and found the ordering affected the results marginally small.

Jinyun Tang 9/5/2016 3:39 PM

Jinyun Tang 9/5/2016 3:42 PM

Jinyun Tang 9/5/2016 3:43 PM

Jinyun Tang 9/5/2016 7:00 PM

Jinyun Tang 9/5/2016 7:00 PM

Jinyun Tang 9/5/2016 3:45 PM

Jinyun Tang 9/12/2016 1:36 PM

Jinyun Tang 9/12/2016 1:37 PM

Jinyun Tang 9/9/2016 12:37 PM

Jinyun Tang 9/5/2016 7:22 PM

Jinyun Tang 9/5/2016 7:16 PM

Jinyun Tang 9/5/2016 7:17 PM

Jinyun Tang 9/5/2016 7:16 PM

Jinyun Tang 9/5/2016 7:18 PM

Jinyun Tang 9/5/2016 7:24 PM
Deleted: To further simplify the problem, we solved the overall spatiotemporal evolution of substrate $S$ (which a function of both transport and biogeochemistry) using the operator splitting approach (e.g., Strang, 1968; Tang et al., 2013), so that $F_{S,input}$ and $F_{S,uptake}$ refer, respectively, to nitrogen mineralization (by decomposers) and nitrogen immobilization (by microbes and plants).

When solved with the forward Euler scheme (e.g. Atkinson, 1989), Equation (1) may be approximated as:

$$S(t + \Delta t) = S(t) + \left(F_{S,input} - F_{S,uptake}\right)\Delta t \qquad (2)$$

With a given numerical time step $\Delta t$, if $S(t + \Delta t)$ becomes negative (before any adjustment to the rates that change $S(t)$), the biogeochemical system is defined as substrate-$S$ limited during that numerical time step. Here we once again remind readers not to confuse this definition (of growth limiting substrate) with using different analytical formulations of how nutrients could limit or co-limit the biogeochemical system (i.e. growth-controlling substrates), because this numerical limitation (i.e. growth limiting substrate) will always occur for whatever analytical formulation (of growth controlling substrates) being used. We also note that this numerical definition of nitrogen limitation (which operates on time scales from minutes to hours) appears different from the ecological definition, which is defined as stimulated ecosystem productivity in response to nitrogen addition and operates on time scales from days to years (Vitousek and Howarth, 1991). However, in a BGC model, ecological nitrogen limitation is realized as an emergent response accumulated from many within time-step nitrogen limitations (and should be considered as a combination of growth-controlling and growth-limiting processes).

Using a higher order numerical scheme will not avoid this numerical substrate limitation, and, when substrate limitation occurs, the high order scheme will usually become first order (Bolley and Crouzeix, 1978), a result that also holds for implicit schemes (Hundsdorfer and Verwer, 2003). Higher order accuracy may be achieved if both the substrate production and destruction rates are modified simultaneously (e.g., Burchard et al., 2003), but such an approach will fail when substrate production is independent of consumption, a situation that occurs exactly in the CENTURY-like soil biogeochemical models (Parton et al. 1988; Koven et al. 2013), where, because the different soil organic matter pools are decayed in a linearly dependent manner, the activity of nitrogen mineralizers is independent from that of nitrogen immobilizers (Tang and Riley, 2016). Nor will an adaptive time stepping approach resolve this numerical substrate limitation problem, because in many cases it would require an impractically small time-step to avoid negative numerical solutions (Formaggia and Scott, 2011). Nevertheless, a numerical nitrogen limitation as applied in equation (2) does depend on the time step size, but as we demonstrate below, this uncertainty is secondary to that from using different numerical implementations of the supply-demand ansatz based nitrogen limitation.

We now analyse three legitimate and commonly applied numerical methods to resolve substrate limitation when solving equation (2). We reveal that the three numerical approaches imply different coupling between nitrogen competitors and producers in the model, they therefore lead to different (sometimes unacknowledged) ecological coupling between carbon and nitrogen dynamics.

The first nitrogen uptake limitation approach has been adopted by models like CLM-CNP (Yang et al., 2014), BiomeBGC (Thornton et al., 2002), BiomeBGC MuSo (Hidy and Barcza, 2014), JSBACH-CN (Parida, 2011; with

Jinyun Tang 9/9/2016 12:43 PM

Jinyun Tang 9/9/2016 12:52 PM

Jinyun Tang 9/5/2016 7:44 PM

Jinyun Tang 9/5/2016 7:44 PM

Jinyun Tang 9/12/2016 1:46 PM

denitrification excluded from $F_{S,uptake}$). CLM4.0 (Oleson et al., 2010), CLM4.5 (Oleson et al., 2013), and one version of ALMv1 (the land model in the DOE earth system model ACME-v1). Mathematically, it reads

$$\bar{\bar{F}}_{S,uptake} = \min \left\{ \frac{S(t)/\Delta t}{F_{S,uptake}}, 1 \right\} F_{S,uptake} \qquad (3)$$

Equation (3) assumes that the actual total nitrogen uptake $\bar{\bar{F}}_{S,uptake}$ is limited solely by the available mineral nitrogen $S(t)$

[revised manuscript text omitted]

Jinyun Tang 9/7/2016 5:25 AM

Jinyun Tang 9/7/2016 5:25 AM

Jinyun Tang 9/7/2016 5:25 AM

Jinyun Tang 9/7/2016 5:17 AM

Jinyun Tang 9/7/2016 5:17 AM

Jinyun Tang 9/7/2016 5:17 AM

Jinyun Tang 9/7/2016 5:18 AM

simulations by adding a nitrogen storage pool to fulfil the nitrogen demand from GPP and refill the nitrogen storage pool through plant uptake (in presence of microbial competition).

ALMv0/CLM4.5BGC employs a fixed CN stoichiometry for plants and a CENTURY-like (Parton et al., 1988) formulation for soil BGC, where the latter represents microbial population dynamics and associated biogeochemical activity implicitly. All models used here allow plants and microbes to compete equally (or proportionally) for $NH_4^+$ and $NO_3^-$, and assume that both plants and organic matter decomposers assimilate $NH_4^+$ over $NO_3^-$. The first assumption (on whether the uptake of $NH_4^+$ and $NO_3^-$ is proportional to their pool sizes) is now under intense debate (e.g., Gerber et al., 2010; Zaehle and Friend, 2010; Thomas et al., 2015; Niu et al., 2016; Zhu et al., 2016), whereas the second assumption is very likely unrealistic because (1) it restricts the model to execute nitrogen limitation after oxygen limitation (as $NO_3^-$ demand by denitrifiers is a function of oxygen and applying nitrogen limitation requires knowing the relative uptake demand of $NH_4^+$ over $NO_3^-$), even though they occur simultaneously in the real world and (2) a grid cell in any large scale BGC model actually represents the average of a heterogeneous soil, so the uptake of $NO_3^-$ should never be zero as long as some $NO_3^-$ exists.

To evaluate hypothesis (H1), we used five BGC model configurations implemented in ALMv0-BeTR (Table 1). Among them, the three BGC formulations (MNL, NUL, and PNL) differ in their numerical interpretations of nitrogen limitation. Since all model configurations in BeTR require identical model inputs, we also tested the model sensitivity to initial conditions by comparing PNL with PNLIC, where the latter uses the code base of PNL and initial conditions from the NUL simulation. Simulations PNLIC and NUL are compared to demonstrate the effect of different nitrogen limitation implementations with the same initial conditions. The final model configuration, PNLO, when compared to PNL, illustrates the ordering dependence of substrate limitation (for oxygen and nitrogen). To prevent the model PNLO from crashing (on negative values of oxygen or mineral nitrogen), we first predicted the relative demand for $NH_4^+$ and $NO_3^-$ based on total mineral nitrogen availability, then implemented oxygen limitation on nitrification and decomposition, and finally applied nitrogen limitation to microbes and plants a second time to obtain the corrected nitrogen uptake for plants and microbes. This requirement to apply nitrogen limitation in a predictor-corrector manner is not easily observable from the governing equations of the BGC model and demonstrates (1) that the default ALMv0/CLM4.5BGC model structures of plant-soil nitrogen interactions are problematic and (2) (once more) that numerical implementations of nutrient limitations in ESM land models may imply (sometimes unacknowledged) different ecological dynamics that is not described in the governing equations. We run our global simulations from 1850 to 2300 (see simulation protocol) and compare the output from 2006-2100 to the reported NEE for CMIP5 simulations (Shao et al., 2013) to evaluate H1.

The second hypothesis (H2) is evaluated with four example single gridcell simulations in geographically and climatically distinct locations (Figure 3): (74.67°W, 40.6°N; Eastern U.S.), (26.22°E, 67.7°N; Northern Finland), (50.02°W, 4.88°S; North East Brazil), and (51.5°W, 30.0°S; South Brazil). These gridcells were chosen to illustrate spatial heterogeneity in how time stepping strategies would influence simulated ecosystem carbon dynamics. We adopted the strategy from Tang and Riley (2016) (their appendix D) for adaptive time stepping and designated relevant simulations with PNL-adapt. H2 is evaluated by comparing the effect of adaptive-time stepping to that of using different numerical implementations of nitrogen limitation.

**2.2 Simulation protocol**

All model simulations were first run to preindustrial equilibrium using the spinup protocol in Koven et al. (2013) with the QIAN climate forcing data (cycled for 1948-1972; Qian et al., 2006). The model output by the end of spinup was then used for simulations in the contemporary period 1850-2000 with diagnostic atmospheric $CO_2$ concentrations. The RCP4.5 scenario atmospheric $CO_2$ concentrations (starting from 2006; see Figure S2b) were used together with the cycled QIAN climate forcing for the simulation period 2001-2300. We did not apply the climate anomaly representing future climate change to the RCP simulations; therefore the simulated carbon dynamics over 2001-2300 only represented the effects of changing atmospheric nitrogen deposition (Figure S2a), atmospheric $CO_2$ (Figure S2b), and land use change. We expect that including more uncertainty sources (such as uncertain future climate) will further strengthen the conclusions of our study (e.g. Tang and Zhuang, 2008). We finally note that the decision to run the simulations to 2300 is inspired by Randerson et al. (2015) and is just to push the models to one type of extreme and see if they would behave unexpectedly.

[revised manuscript text omitted]
 (which lead to a stronger increase in heterotrophic respiration than in NPP under more efficient nitrogen uptake; see discussion in section 4.1). We also found that the predicted total soil carbon change is more sensitive than the total vegetation carbon change (Figure 2 and Figure S6) in response to the different nitrogen limitation implementations, indicating stronger nitrogen regulation of soil carbon stocks.

**3.3 Point simulations for the four sites**

For the group of simulations conducted at the four grid points (Figure 3), we observed similar divergences as those in the global simulations (Figure 2): the MNL scheme (blue lines) predicted higher carbon gain than did the PNL scheme (red lines), yet the NUL predictions (green lines) almost overlapped with those by MNL. Invoking adaptive time-stepping (PNL-adapt; magenta lines) further decreased the predicted carbon gain, which could be explained by the even more effective nitrogen uptake implied by the PNL scheme under smaller time steps. We also switched the computing order between reaction and transport for PNL-adapt (which like all simulations reported in this text calculates biogeochemical reaction before transport) and only found negligible difference (Figure S9).

**3.4 Results of hypotheses evaluation**

Taking all simulations together, we conclude that hypothesis H1 is affirmed given the spread of our simulated cumulative land carbon uptake is larger than that in Shao et al. (2013) for CMIP5 models. Meanwhile, H2 is satisfied in some, but not other, sites and that the size of the numerical time step could have either significant (Figure 3a) or secondary (Figure 3b, c, and d) importance on simulated ecosystem carbon stocks trajectories.

**4. Discussion**

Below we first discuss how the three different numerical implementations of nitrogen limitation led to different partitioning of nitrogen fluxes. Then we explore the importance of the coarse woody debris in affecting the simulated spread in land-atmosphere carbon exchange. Finally we give our recommendations on how to alleviate the type of uncertainty we identified in this study.

**4.1 Reasons for the large C cycle differences between different nitrogen limitation implementations**

We observed that PNL, NUL, and MNL schemes predict sequentially stronger nitrogen limitation under the same mineral nitrogen availability (Supplemental Material). For biogeochemical models like ALM that resolve mineral nitrogen into ammonium and nitrate (together with the assumed preference of ammonium over nitrate), this order of limitation translates into sequentially less effective plant and microbial assimilation of ammonium and stronger uptake of nitrate nitrogen. Indeed, we found PNL-adapt predicted the highest nitrification rate (as nitrifiers are competing for ammonium in ALM) followed by PNL and MNL (which overlapped with NUL; see Figure 4a1, b1, c1, a2, b2 and c2), leading to the same ranking of soil nitrate abundance (Figure S10) and nitrate loss through aqueous transport (Figure 4a4, b4, and c4). The difference in denitrification rates as simulated by different nitrogen limitation schemes is also evident, with the lowest value predicted by PNL-adapt, and increasing in MNL (which overlaps NUL) and then PNL. The simulations at 51.5° W, 30.0 ° S (which ALM identifies as a $C_3$ grassland) only qualitatively resemble those at the other three sites, yet the ranking of soil nitrate abundance remains (Figure S10d). Corresponding to the nitrogen dynamics, the ecosystem heterotrophic respiration also increases in the order of MNL (which overlaps NUL), PNL, and then PNL-adapt, except for the period after 1980 for the fourth site (Figure 5d), indicating a strong sensitivity of carbon dynamics to nitrogen processes. For global simulations driven by RCP4.5 atmospheric $CO_2$ forcing over 2001-2300, this stronger increase of respiration led PNL to predict about 3200 Pg C more respiration than did by MNL, yet the NPP predicted by PNL only increased by 1900 Pg C as compared to that by MNL, which together led the PNL scheme to predict a lower $CO_2$ fertilization effect.

**4.2 High sensitivity of coarse woody debris dynamics to nitrogen**

We observed that the response of coarse woody debris (CWD) pool dominated the simulated difference in total land-atmosphere carbon exchange during both the contemporary period 1850-2000 (Figure S3) and the projection period 2001-2300 (Figures S7 and S8). A smaller fraction is contributed from carbon in woody product and seed (see second rows of Figures S3, S7 and S8). In ALMv0/CLM4.5BGC, coarse woody debris is accumulated from mortality due to fire (predicted with the model by Li et al. (2012a, b)) and background death (2% per year; Oleson et al., 2013), harvest and land use change, and it is lost through decomposition into lignin and cellulose. The decomposition of CWD immobilizes nitrogen, and is assumed to produce no $CO_2$, where the latter is obviously contradictory to reality (e.g. Gough et al., 2007). Nevertheless, the more efficient nitrogen uptake as ranked in PNL, NUL and MNL (supplemental material) has led to the

sequentially higher loss of CWD in the order of low to high as MNL, NUL, PNL (PNLO) and PNLIC (Figures S7 and S8). PNLIC has predicted the highest loss of CWD, because it uses the initial condition from NUL and NUL has accumulated more CWD during the spinup period due to its less efficient nitrogen uptake as compared to PNL. Regionally, the tropics showed the largest spread in the predicted CWD loss ($-121\sim214$ g C m$^{-2}$ yr$^{-1}$), followed by the north temperate region ($-156\sim10$ g C m$^{-2}$ yr$^{-1}$), south of 23° S ($-10\sim8$ g C m$^{-2}$ yr$^{-1}$) and the Arctic ($-4\sim2$ g C m$^{-2}$ yr$^{-1}$; see Figure S6 and S7). Such high sensitivity of the CWD dynamics with respect to the nitrogen dynamics further indicates urgency to develop robust implementations of nitrogen limitation in ESM land biogeochemical models.

**4.3 Strategies for robust carbon and nitrogen coupling**

[revised manuscript text omitted]
_1\left(t+\Delta t\right)=N_1\left(t\right)+\Delta t N_1\left(t\right)\left[\min\left\{\frac{g_{11}\left(R_1\left(t\right)\right)}{q_{11}},\frac{g_{12}\left(R_2\left(t\right)\right)}{q_{12}}\right\}-D_1\right] \tag{A-3}$$

$$R_1\left(t+\Delta t\right)=R_1\left(t\right)+\Delta t f_1\left(R_1\right)-\Delta t q_{11}\left[\min_j\left\{\frac{g_{ij}\left(R_j\right)}{q_{ij}}\right\}\right]N_1, j=1,2 \tag{A-4}$$

$$R_2\left(t+\Delta t\right)=R_2\left(t\right)+\Delta t f_2\left(R_2\right)-\Delta t q_{12}\left[\min_j\left\{\frac{g_{ij}\left(R_j\right)}{q_{ij}}\right\}\right]N_1, j=1,2 \tag{A-5}$$

Now suppose population $N_1$ is locally limited by substrate $R_1$, such that $g_{11}\left(R_1\left(t\right)\right)/q_{11}<g_{12}\left(R_2\left(t\right)\right)/q_{12}$, which leads to

$$N_1\left(t+\Delta t\right)=N_1\left(t\right)+\Delta t N_1\left(t\right)\left[\frac{g_{11}\left(R_1\left(t\right)\right)}{q_{11}}-D_1\right] \tag{A-6}$$

$$R_1\left(t+\Delta t\right)=R_1\left(t\right)+\Delta t\left[f_1\left(R_1\right)-g_{11}\left(R_1\right)N_1\right] \tag{A-7}$$

$$R_2\left(t+\Delta t\right)=R_2\left(t\right)+\Delta t\left[f_2\left(R_2\right)-\frac{q_{12}}{q_{11}}g_{11}\left(R_1\right)N_1\right] \tag{A-8}$$

Now define

$$\lambda=\frac{g_{11}\left(R_1\left(t\right)\right)}{g_{12}\left(R_2\left(t\right)\right)}\frac{q_{12}}{q_{11}} \tag{A-9}$$

Where it can be verified that $\lambda<1$. Then by entering equation (A-9) into (A-8), we obtain

$$R_2\left(t+\Delta t\right)=R_2\left(t\right)+\Delta t\left[f_2\left(R_2\right)-\lambda g_{12}\left(R_2\right)N_1\right] \tag{A-10}$$

Now if $R_1(t+\Delta t)>0$ and $R_2(t+\Delta t)<0$, both of which can be easily satisfied (note $f_2(R_2)$ could be negative), then population $N_1$ is de facto limited by substrate $R_2$, which is opposite to the "local constraint" that the growth of population $N_1$ is limited by substrate $R_1$. Now in order to avoid $R_2(t+\Delta t)<0$, a numerical substrate limitation must be done, and the use of Liebig's law of minimum in growth rate calculation in equation (A-3) is inappropriate such that it results in a double counting of substrate limitation. For a community of many populations and substrate, we expect such misuse of Liebig's law of minimum could occur even more frequently, and should be avoided.

**Author Contributions**

J.Y. Tang designed the study and conducted the experiments. J.Y. Tang and W. J. Riley discussed the results and wrote the paper.

**Acknowledgments**

This research was supported by the Director, Office of Science, Office of Biological and Environmental Research of the US Department of Energy under contract No. DE-AC02-05CH11231 as part of the Accelerated Climate Model for Energy in the Earth system Modeling program, as well as the Regional and Global Climate Modeling (RGCM) Program. The study used the Lawrencium computational cluster resource provided by the IT Division at the Lawrence Berkeley National Laboratory. The data used in this paper can be obtained by contacting the first author at jinyuntang@lbl.gov.

Table 1. Model configurations used to evaluate the uncertainty of ambiguous numerical implementation of nutrient limitation.

| Simulation ID | Model configuration |
|---|---|
| MNL | Mineral Nitrogen based Limitation scheme: only existing mineral nitrogen is available for uptake at current time step. It implements equation (3). |
| NUL | Net nitrogen Uptake based Limitation scheme: mineral nitrogen demand is calculated as the residual between total nitrogen demand and gross mineralization. It implements equation (6). |
| PNL | Proportional Nitrogen flux based limitation scheme: mineral nitrogen from gross mineralization and existing soil mineral nitrogen are competed equally by plants and microbes. It implements equation (7). |
| PNLIC | Like PNL, but it uses initial condition from NUL. |
| PNLO | Like PNL, but $O_2$ limitation comes after nitrogen limitation. However, a second nitrogen limitation required for avoiding model crash. |
| Default | ALMv0, which is the de facto CLM4.5BGC. |

Jinyun Tang 9/13/2016 1:34 PM

Jinyun Tang 9/9/2016 2:42 PM

[Figure]

Figure 1. Model predictions for the contemporary period 1850-2000: (a) Cumulative net ecosystem exchange (NEE; positive into the atmosphere); (b) Gross primary productivity; (c) Net primary productivity; (d) July leaf area index; (e) July latent heat flux; (f) total organic soil carbon to 1 m depth; (g) total organic soil nitrogen to 1 m depth; (h) total vegetation carbon; and (i) total vegetation nitrogen. Results for (b)-(i) are averaged over 1991-2000.

[Figure]

Figure 2. Model simulations forced by the Representative Concentration Pathway 4.5 (RCP4.5) atmospheric $CO_2$ for year 2001-2300. Here total soil carbon includes litter carbon and soil organic matter as defined in CLM4.5 (Oleson et al., 2013); coarse woody debris is excluded (but can be found in Figure S7). All changes are calculated as relative to each of their initial carbon pool sizes at the start of the simulation (year 2000). The decadal oscillation shown in the figure is due to the cycling of the QIAN climate forcing.

Jinyun Tang 9/12/2016 4:20 PM

[Figure]

Figure 3. Point simulations for the 4 specific gridcells using different model configratuions. For each site, all simulations used identical initial conditions obtained from spinup with the PNL-adapt code. Note the color schemes are different from that in Figure 1 and Figure 2. The decadal oscillation shown in the figure is due to the cycling of the QIAN climate forcing.

[Figure]

Figure 4. Nitrogen fluxes for the four specific gridcell simulations obtained from different model configurations. The four columns from left to right correspond to the four locations specified in Figure 3. The decadal oscillation shown in the figure is due to the cycling of the QIAN climate forcing.

[Figure]

Figure 5. Heterotrophic respiraiton for the four specific gridcell simulations obtained from running different model configurations. The decadal oscillation shown in the figure is due to the cycling of the QIAN climate forcing.

Jinyun Tang 9/11/2016 2:26 PM

Supplemental materials for "**Potentially large uncertainty in ecosystem carbon dynamics resulting from ambiguous numerical coupling of carbon and nitrogen biogeochemistry: a demonstration with the ACME land model**"

**Jinyun Tang and William J. Riley**

Earth and Environmental Science Area, Lawrence Berkeley National Lab (LBL),

Berkeley, CA, United States

Corresponding to: J.Y. Tang, jinyuntang@lbl.gov

**1. Introduction of contents**

The first part of this supplemental material reports the proof of the sequentially weaker nitrogen limitation in the application of the MNL, NUL and PNL numerical nitrogen limitation schemes. The second part contains figures (**S1-S11**) that provide complementary information to support our results and conclusions in the main text.

**2. Proof of the progressively weaker nitrogen limitation**

We prove under the same soil mineral nitrogen availability and fluxes of $F_{S,input}$ and $F_{S,uptake}$ that the application of MNL, NUL and PNL schemes leads to progressively weaker nitrogen limitation.

We first prove $\overline{F}_{MNL,uptake} < \overline{F}_{NUL,uptake}$, where, without confusing the readers, the subscript $S$ was removed.

Because substrate $S$ is limited, $\overline{F}_{MNL,uptake} < \overline{F}_{NUL,uptake}$ is equivalent to

$$\frac{S(t)/\Delta t}{F_{S,uptake}} < \frac{S(t)/\Delta t}{F_{S,uptake} - F_{S,input}} \qquad (S\text{-}1)$$

which is reduced to $F_{S,uptake} - F_{S,input} < F_{S,uptake}$, a condition always holds.

We now prove $\overline{F}_{NUL,uptake} < \overline{F}_{PNL,uptake}$. This requires

$$\frac{S(t)/\Delta t}{F_{S,uptake} - F_{S,input}} < \frac{F_{S,input} + S(t)/\Delta t}{F_{S,uptake}} \tag{S-2}$$

By rearranging the terms of (S-2), we have to show

$$F_{S,uptake} S(t)/\Delta t < \left(F_{S,input} + S(t)/\Delta t\right)\left(F_{S,uptake} - F_{S,input}\right) \tag{S-3}$$

which after some rearrangement becomes

$$S(t)/\Delta t < \left(F_{S,uptake} - F_{S,input}\right) \tag{S-4}$$

Since (S-4) is the definition of substrate limitation for the NUL scheme, it always holds under substrate limitation.

We now finish our proof.

**List of supplemental figures**

[Figure]

Figure S1. A demonstration of the tracer transport accuracy of BeTR. The Hydro water is water simulated with the biophysics module in the ACME land model. BeTR water is water tracer tracked in BeTR. Ideally, the linear fit should be one to one.

[Figure]

Figure S2: (a) Cumulative atmospheric deposition from 1850 to 2300. (b) Atmospheric $CO_2$ from 1850 to 2300. The small zigzag in (b) is due to that RCP4.5 $CO_2$ starts from 2006.

[Figure]

Figure S3: (a1), (b1) and (c1) are carbon changes in total coarse woody debris. (a2), (b2) and (c2) are changes in total product carbon and seed carbon.

[Figure]

Figure S4. Simulated cumulative carbon fluxes in the contemporary period 1850-2000.

[Figure]

Figure S5: Latitudinal distribution of simulated soil mineral nitrogen for 1991-2000. (a) Total soil mineral nitrogen; (b) $NH_4^+$ and (c) $NO_3^-$.

[Figure]

Figure S6: Model simulations for the scenario Representative Concentration Pathway 4.5 (RCP4.5) atmospheric $CO_2$ for the years 2001-2300. Here total soil carbon includes litter carbon and soil organic matter as defined in CLM4.5; coarse woody debris is excluded. All changes are calculated as relative to each of their initial carbon pool sizes at the start of the simulation (i.e. end of year 2000). The oscillations as shown in the figure are due to the cycling of the QIAN climate forcing.

[Figure]

Figure S7: Simulated evolution of coarse woody debris carbon (a1-c1) and product and seed carbon (a2-c2) for the RCP 4.5 $CO_2$ driven period 2001-2300. These results are complementary to Figure 2 in the main text.

[Figure]

Figure S8: Simulated evolution of coarse woody debris carbon (a1-c1) and product and seed carbon (a2-c2) for the RCP 4.5 $CO_2$ driven period 2001-2300. These results are complementary to Figure S6 above.

[Figure]

Figure S9: Evaluation of the ordering effect for the point simulations. PNL-adapt-tr simulates transports ahead of biogeochemical calculations, whereas PNL-adapt does the opposite order. From left to right, the four columns are representing sites that are corresponding to the locations specified in Figure 3 of the main text.

[Figure]

Figure S10: Soil nitrate concentrations for the point simulations as obtained from different model configurations.

[Figure]

Figure S11: A demonstration of the zigzag phenomena and the strong time-stepping dependence of the numerical solution using Euler methods.